# The F1148 hydrophobic lock: A critical determinant of SARS-CoV-2 spike protein-mediated membrane fusion via the 3H/CH cavity

Fuzhi Lei[1], Yahan Lei[1], Zhenghong Yuan[1]*, Zhigang Yi[1,2]*

1 Key Laboratory of Medical Molecular Virology (MOE/NHC/CAMS), Shanghai Institute of Infectious Disease and Biosecurity, Shanghai Frontiers Science Center of Pathogenic Microorganisms and Infection, School of Basic Medical Sciences, Fudan University, Shanghai, China, 2 Shanghai public health clinical center, Fudan University, Shanghai, China

* zhyuan@shmu.edu.cn (ZY); zgyi@fudan.edu.cn (ZY)

## Abstract

The S2 subunit of the coronavirus Spike protein undergoes extensive conformational refolding to drive membrane fusion during viral entry. Although the HR1/HR2 six-helix bundle (6-HB) is recognized as the core mediator of fusion, the molecular driving force governing its formation remains poorly elucidated. Here, through systematic mutagenesis of the AlphaFold-predicted stem helix (SH) region in S2, followed by analysis of the resulting SC2-VLP entry phenotypes, we identified key amino acid residues within conserved helices that are present in both prefusion and postfusion Spike conformations. These elements, which we term postfusion-preserved helices (PFPHs), were found to be critical for SC2-VLP entry. Structural analysis revealed a "hydrolock" interaction between F1148 in PFPH-1 and a conserved cavity formed by 3H (I742, C749)/CH (I993, L996, I997). Deep mutational scanning demonstrated that only hydrophobic residues at F1148 were functionally viable and essential for membrane fusion, underscoring the critical role of a hydrophobic lock ("hydrolock") interaction between F1148 and the 3H/CH cavity in membrane fusion. Furthermore, HA-replacement mutagenesis and anti-HA neutralization assays showed that significant neutralization activity was restricted to HA insertions proximal to PFPH-1, selectively inhibiting membrane fusion without affecting receptor binding. Notably, the 3H/CH cavity remains structurally stable across Spike conformations, being sequentially occupied by prefusion-L977, intermediate-F782, and postfusion-F1148. We propose a model wherein hydrolock interactions drive S2 refolding and fusion by displacing intermediate interactions. This study provides mechanistic insights into Spike dynamics and highlights hydrolock interactions as a promising target for broad-spectrum antiviral strategies.

**Data availability statement:** All relevant data are within the manuscript and its Supporting Information files.

**Funding:** This work was in part supported by the Shanghai Municipal Science and Technology Major Project (ZD2021CY001 to ZHY [Zhenghong Yuan]) and the Shanghai Municipal Health Commission (GWV1-11.2-XD30 to ZGY [Zhigang Yi]). The funders had no role in study design, data collection and analysis, decision to publish, or preparation of the manuscript.

**Competing interests:** The authors have declared that no competing interests exist.

## Author summary

The S2 subunit of coronavirus Spike protein is highly conserved and indispensable for viral entry, representing an attractive target for broad-spectrum antiviral strategies. Through a combination of mutagenesis and structural analyses, we identified a conserved hydrophobic interaction—termed the "hydrolock"—between the F1148 residue within the newly defined postfusion-preserved helix (PFPH-1) motif and a structurally conserved 3H/CH cavity, which serves as a critical determinant for membrane fusion. Moreover, we demonstrated that significant neutralizing activity is confined to regions adjacent to PFPH-1, achieved by inhibiting the membrane fusion process. We propose a mechanistic model where hydrolock interactions orchestrate the displacement of intermediate interactions, thereby facilitating S2 refolding and membrane fusion. These findings deepen our understanding of Spike protein dynamics and provide a foundation for developing next-generation antiviral strategies targeting this conserved mechanism.

## Introduction

Coronavirus species in the *Coronaviridae* are important emerging and re-emerging pathogens. Members of the *betacoronavirus* genus such as severe acute respiratory syndrome coronavirus (SARS-CoV) [1,2], middle east respiratory syndrome coronavirus (MERS-CoV) [3] and severe acute respiratory syndrome coronavirus 2 (SARS-CoV-2) [4,5] can lead to either pandemics or severe diseases. Despite the worldwide advancement and implementation of diverse vaccines, therapeutic antibodies, and peptides, the persistent mutations of the virus pose significant challenges to effective control measures [6–8]. Consequently, the development of broad-spectrum antiviral strategies against Coronavirus remains of great importance.

Coronavirus are a single-stranded positive-sense RNA virus, whose genomic RNA functions directly as messenger RNA for translation of ORF1a and ORF1ab, the latter of which is produced by a programmed ribosomal frameshift. ORF1a and ORF1ab are processed by viral proteases to generate non-structural proteins for viral transcription and replication. Transcribed viral subgenomic RNAs act as messenger RNAs for structural proteins for virion assembly [9]. Coronavirus virion comprises four structural proteins: Spike (S), Membrane (M), Envelope (E), and Nucleocapsid (N) [10]. The Spike protein is a highly glycosylated, trimeric type I transmembrane protein composed of two distinct subunits, S1 and S2 [11,12]. The S1 subunit mediates attachment to host cells and comprises the N-terminal domain (NTD), receptor-binding domain (RBD), C-terminal domain 1 (CTD1), and C-terminal domain 2 (CTD2). The S2 subunit is responsible for the fusion of the viral membrane with the host cell membrane after receptor binding-mediated S1 shedding. It consists of the three-helix segment (3H), fusion peptide proximal region (FPPR), fusion peptide (FP), heptad repeat 1 (HR1), central helix region (CH), connector domain (CD), heptad repeat 2 (HR2), the transmembrane anchor (TM), and the cytoplasmic tail (CT) [11,13]. While the S1 subunit is subject to host immune pressure and exhibits high variability, the S2 subunit remains relatively conserved [14].

After furin cleavage, the S1 and S2 subunits are linked by non-covalent bonds, forming a metastable state that promotes subsequent conformational changes [15–17]. Spike exhibiting a "cloverleaf" shape in its prefusion conformation, the "up" RBD engages with the Angiotensin-Converting Enzyme 2 (ACE2) receptor, promoting stable attachment to the cell surface [17,18]. Following this interaction, Spike is cleaved by proteases such as Transmembrane Serine Protease 2 (TMPRSS2) on the cell surface or Cathepsin L intracellularly, leading to the detachment of the S1 subunit [19–21], which exposes the fusion peptide. Subsequently, the S2 subunit undergoes substantial conformational changes, with the highly hydrophobic FP extending and integrating into the target host cell membrane, resulting in an unstable "prehairpin intermediate" structure [22]. Concurrently, the trimeric helical bundle of the prehairpin intermediate loosens to facilitate S2 refolding, which in turn yields a stable "rod" shaped postfusion conformation. This refolding process approximates the viral membrane to the host membrane, allowing for the membrane fusion and release of the viral genome into the host cell [23–25].

The structures of prefusion and postfusion conformations with high resolution have been resolved. However, determining the structure of the "prehairpin intermediate" conformation with high resolution remains challenging, as it is believed to be short-lived [22]. Two excellent studies employed systems based on the surface expression of Spike and ACE2 on particle membranes, utilizing lipopeptide entry inhibitors or antibodies to block the membrane fusion process, thereby trapping Spike in a refolding intermediate conformational state. These studies successfully detected the EM density corresponding to this conformation; however, limitations in resolution have precluded detailed structural analysis [22,26]. A recent study utilizing spike proteins lacking the transmembrane domain and soluble ACE2, captured an intermediate in the early stage of membrane fusion, where HR1 and FP have already protruded, while the S2 subunit has yet to undergo refolding [27]. Additionally, it has been proposed that glycan-mediated steric hindrance delays S2 refolding to facilitate the fusion peptides to capture the host cell membrane [28], while the precise triggers for this process remain undefined.

In the prefusion conformation of the Spike protein, a stem helix (SH) region is present. In the postfusion conformation, this SH region encompasses the HR2 domain, which pairs with HR1 to form the six-helix bundle (6-HB)—the core structural motif that drives membrane fusion. Notably, peptides designed to target this 6-HB structure have shown potent antiviral efficacy [29,30]. However, whether other critical domains or residues within the SH impact the membrane fusion process remains unclear. Several neutralizing antibodies targeting distinct sites within the SH have been reported to exhibit broad-spectrum activity against SARS-CoV-2 variants and other coronaviruses, including MERS-CoV and HCoV-OC43 [31–34], with a mechanism of inhibition of S2 refolding during membrane fusion [26,31,35]. Despite these advances, the molecular mechanism by which S2 undergoes conformational rearrangements to drive membrane fusion and form the postfusion hexamer remains to be fully elucidated.

In this study, to gain mechanistic insights into SH-mediated S2 refolding and membrane fusion, we performed a mutagenesis analysis of the AlphaFold-predicted stem helix (SH) region and identified critical amino acids within the designated postfusion-preserved helix (PFPH) for membrane fusion. Notably, we demonstrate that the hydrophobic lock (hydrolock) interaction between the hydrophobic residue at position F1148 and the hydrophobic cavity formed by the 3H (I742, C749)/CH (I993, L996 and I997) region is indispensable for membrane fusion. Furthermore, an HA-replacement mutagenesis combined with anti-HA neutralization assays revealed that significant neutralization occurred only when HA was inserted near the F1148 region, resulting in inhibition of membrane fusion but not receptor binding. Based on these findings, we proposed a model in which hydrolock interactions between F1148 and the 3H/CH cavity drive S2 refolding necessary for membrane fusion.

## Results

### Prediction of the SARS-CoV-2 Spike SH trimer

The resolved structure of prefusion and membrane-bound postfusion SARS-CoV-2 spike containing the NTD, RBD, CTD1, CTD2, 3H, FPPR, FP, HR1, CH, CD, HR2, TM, CT domains and SH region spanning residues 1140–1162 have

been characterized [13,24] (Fig 1A). The structure of the region spanning residues 1163–1273 in prefusion conformation is absent. To address this gap, we employed AlphaFold 3 to predict the structure of the Spike protein region spanning residues 1135–1273 (S1 Data). Structural alignment of this predicted model with the partially resolved prefusion Spike structure (PDB ID: 6XR8) (residues 1–1162) showed a rod-like triple-helix conformation, which yielded a root mean square deviation (RMSD) of 0.795 Å (Fig 1B), in agreement with the structure generated by all-atom molecular dynamics simulations [12,28]. In the alignment structure, the triple-helix begins at residue P1140 and extends to K1211, excluding the TM domain. Therefore, we define the SH the region as encompassing residues 1140–1211 (Fig 1B). In contrast, in the postfusion Spike structure, most of the SH region transitions into random coil configurations [13], with only three helices remaining. We designated these conserved elements as postfusion-preserved helix (PFPH-1; residues 1147–1153), PFPH-2 (residues 1168–1172) and PFPH-3 (residues 1179–1193) (Fig 1C). We hypothesized that these PFPHs play an important role in membrane fusion during the Spike transition from prefusion to postfusion. Notably, the SH region, and particularly the PFPHs, are highly conserved across betacoronaviruses (Fig 1D).

## Optimization of the SARS-CoV-2 VLP-based pseudoparticle (SC2-VLP) system

To dissect the roles of the SH region, particularly the PFPHs in membrane fusion, we first optimized a SARS-CoV-2 VLP-based pseudoparticle (SC2-VLP) [36] (S1A Fig). The SC2-VLP supports single-round infection of HEK293T cells co-expressing ACE2 and TMPRSS2, with infection efficiency quantified by measuring firefly luciferase (Fluc) reporter activity in target cells [36] (S1A Fig). Using a uniform design strategy [37,38], we optimized the transfection ratio of the S, M/E, N and Fluc-PS9 plasmids, resulting in an approximately fourfold increase in infection signal compared to previously reported conditions [36] (S1B Fig).Omission of any structural proteins completely abolished the SC2-VLP infectivity, which remained strictly dependent on ACE2 and TMPRSS2 expression (S1C Fig and S2 Table). Western blot analysis of methanol-precipitated supernatants confirmed that Spike secretion required co-expression of N, M, and E, indicating efficient incorporation into SC2-VLPs (S2A Fig). We further introduced an HA-tag into the N-terminal (NT) region of M protein, which had negligible effects on SC2-VLP infectivity (Figs S2B and S2C). Co-immunoprecipitation with $M^{HA}$ confirmed the assembly of all structural proteins (S, E, and N), whereas non-tagged M failed to pull down these components (S2D Fig). Therefore, in subsequent experiments, Spike secretion measured via methanol precipitation was used as an indicator of Spike assembly into SC2-VLPs.

To enable efficient detection of the Spike protein, we evaluated the insertion of an HA-tag within the Spike NTD. Insertion of the HA-tag after Q23, I68, or H245 had minimal impact on SC2-VLPs infectivity (S2E and S2F Figs). Western blot analysis using an anti-S2 antibody confirmed that these insertions did not significantly alter Spike incorporation into SC2-VLP (IB: S2) (S2G Fig). Among the tested variants, HA insertion after Q23 provided the highest detection sensitivity with the anti-HA antibody (IB: HA) (S2G Fig). Therefore, the Spike·(23-NTD$^{HA}$) construct was selected for subsequent mutagenesis experiments.

## Mutagenesis of the Spike SH by Alanine scanning

Using the optimized SC2-VLP system, we performed a sequential triple-alanine scanning mutagenesis across the SH region (residues 1140–1211) and its N-terminal extension (residues 1123–1139) in the HA-tagged Spike construct Spike·(23-NTD$^{HA}$). The resulting SC2-VLP-(S·Mut) variants were co-transfected with N, M/E and Fluc-PS9 plasmids (Fig 2A). Supernatants from transfected cells were collected and used to infect HEK293T cells co-expressing ACE2 and TMPRSS2. Infection assay results showed that mutations Mut 2 (residues 1126–1128) and Mut 3 (residues 1129–1131) in the N-terminal extension of the SH, and Mut 9 (residues 1147–1149) within the PFPH-1 domain, reduced SC2-VLP infectivity by more than 100-fold compared with wild type (WT) (Fig 2B and 2C). In addition, Mut 22 (residues 1185–1187) within the PFPH-3 domain decreased SC2-VLP infectivity by over 10-fold (Fig 2B and 2C). The residues corresponding to Mut22 in PFPH-3 contribute to the formation of the HR1/HR2 six-helix bundle (6-HB), a structural core essential for driving membrane fusion [29,30].

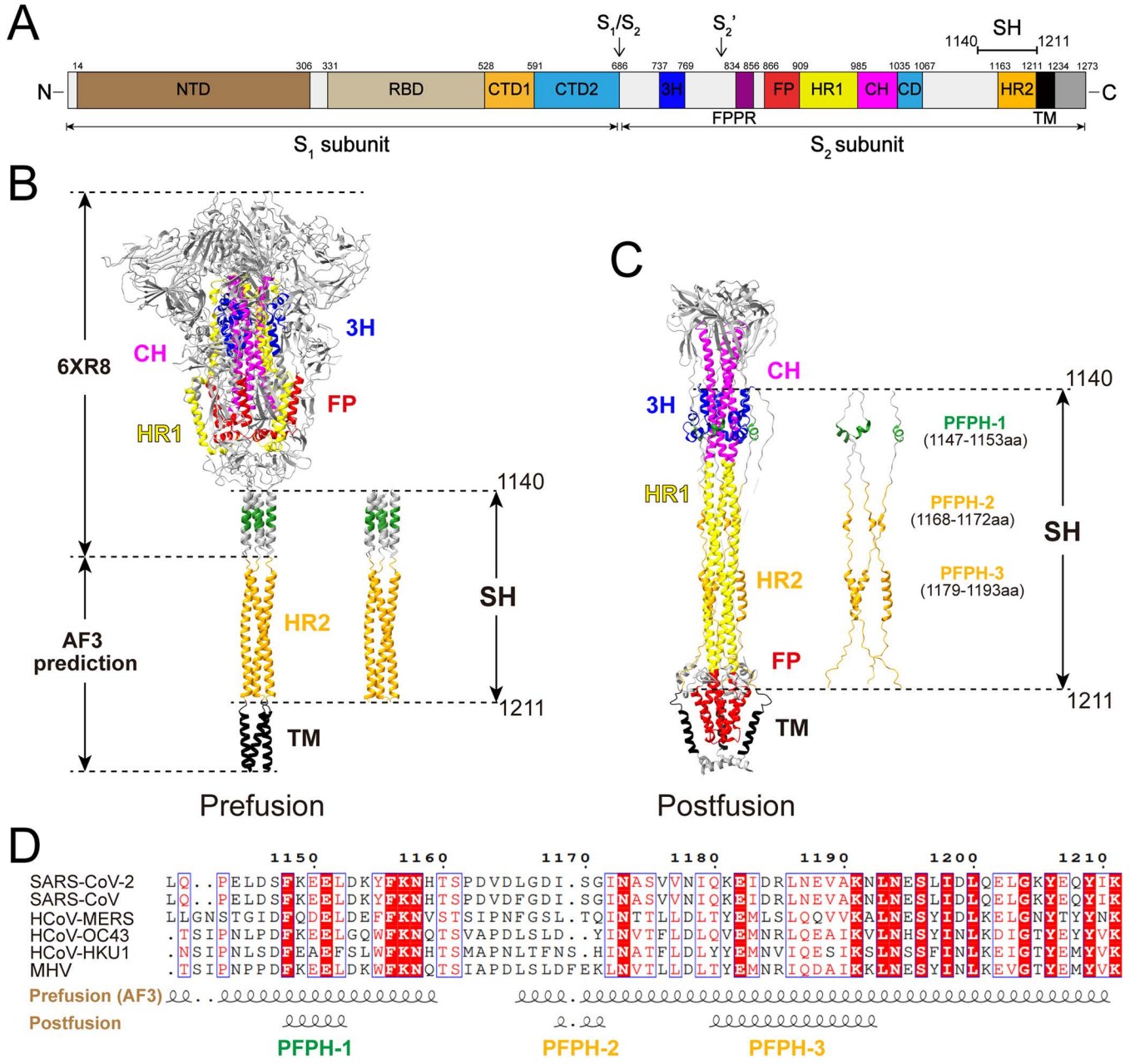

**Fig 1. Prediction of the SARS-CoV-2 Spike SH trimer. (A)** Schematic representation of SARS-CoV-2 Spike protein. Key structural domains are labeled: The NTD (N-terminal domain), RBD (Receptor-binding domain), CTD1(C-terminal domain 1), CTD2 (C-terminal domain 2), S1/S2 cleavage site, 3H (Three-helix segment), S2' cleavage site, FPPR (Fusion peptide proximal region), FP (Fusion peptide), HR1 (Heptad repeat 1), CH (Central helix region), CD (Connector domain), HR2 (Heptad repeat 2), TM (Transmembrane helix) and CT (Cytoplasmic tail). **(B)** Prefusion conformation structure of Spike. The image was generated by aligning PDB 6XR8 with the AlphaFold3-predicted SH trimer. **(C)** Postfusion conformation structure of Spike. The image was generated from PDB 8FDW. Color scheme: PFPH-1 (forest green), PFPH-2 (orange), PFPH-3 (orange), 3H (blue), CH (magenta), HR1 (yellow), HR2 (orange), FP (red), TM (black). **(D)** Multiple sequence alignment of betacoronavirus Spike SH regions performed using Clustal Omega and visualized with ESPript 3.0. Identical residues (100% conservation) are highlighted in red, while conserved residues (≥70% identity based on ESPript 3.0 parameters) are boxed in blue with red text.

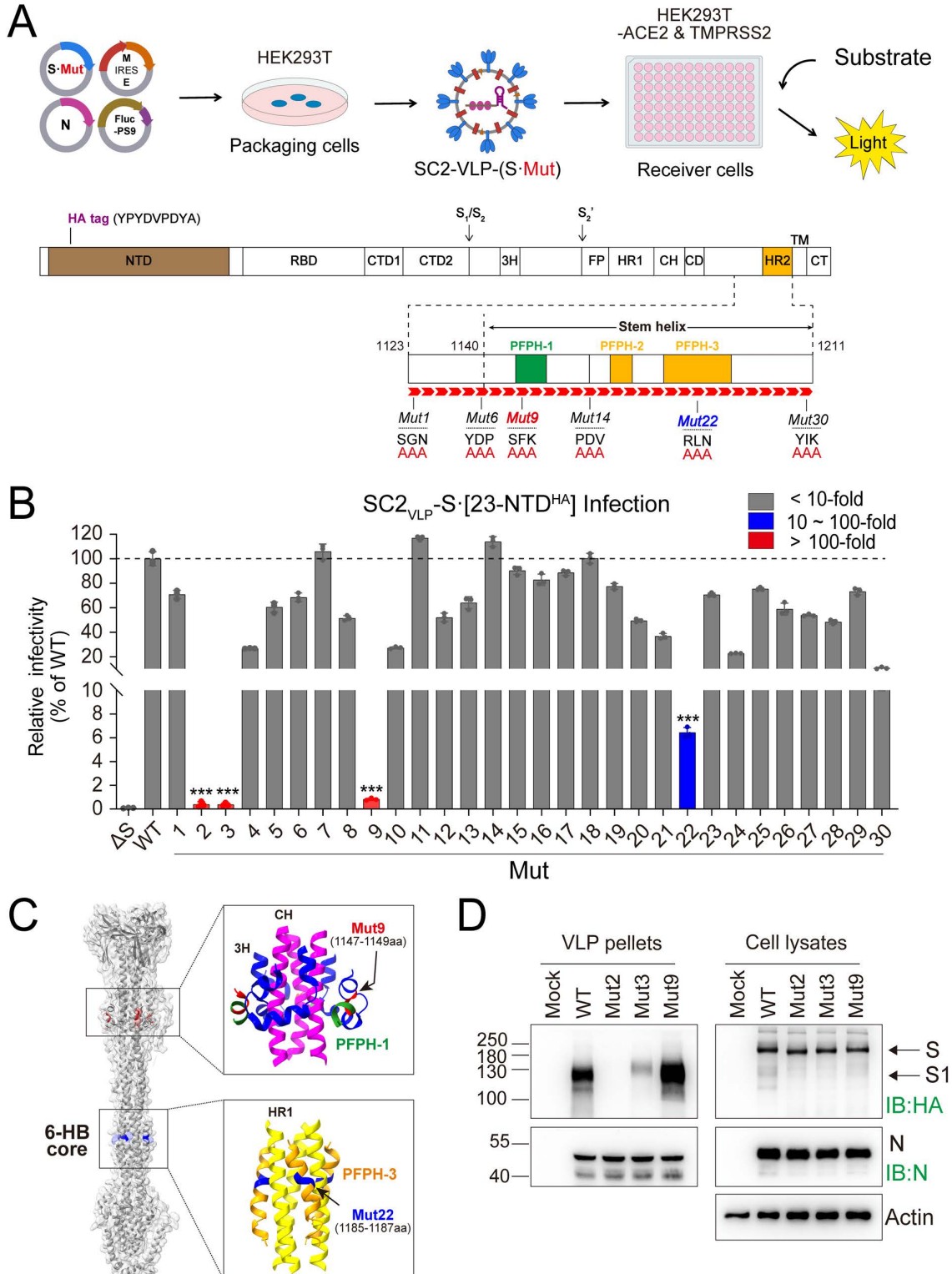

**Fig 2. Alanine scanning mutagenesis of Spike SH region. (A)** Experimental schematic of the Spike mutagenesis. SC2-VLP was generated by co-transfection plasmids expressing Spike **(S)**, membrane **(M)**, envelope **(E)**, nucleocapsid **(N)** and Fluc-PS9. Triple alanine substitutions (Ala-Ala-Ala) (Red arrowheads) were introduced in specified SH regions. An HA-tag was inserted after residue Q23 in the NTD domain. **(B)** Infectivity of SC2-VLPs

in HEK293T-ACE2&TMPRSS2 cells. The intracellular Firefly luciferase (Fluc) activity was determined 24 hours post-infection (mean values ± SDs, n = 3, ***P < 0.001; two-tailed, unpaired t-test). **(C)** Structural localization of Mut9 and Mut22 in Spike postfusion conformation (PDB ID: 8FDW), the structure is shown in surface display mode. **(D)** Western blot analysis of SC2-VLP. The SC2-VLP-containing supernatants precipitated by methanol (pellets) and the packaging cell lysates were subjected to Western blotting assay with the indicated antibodies (IB). The values to the left of the blots are molecular sizes in kilodaltons. Mock: untransfected HEK293T control. The representative images shown for each group from multiple independent experiments.

To determine whether the observed reduction of SC2-VLP infectivity resulted from impaired virion assembly or defective viral entry, we performed Western blot analysis of the SC2-VLP precipitated from culture supernatants. Compared with the WT, Mut 2 and Mut 3 markedly impaired Spike incorporation into SC2-VLP without altering intracellular Spike levels. In contrast, Mut 9 did not affect either Spike incorporation into SC2-VLP or intracellular expression (Fig 2D), suggesting that Mut9 specifically impairs the viral entry process.

### Identification of a hydrophobic lock (hydrolock) interaction formed by F1148/3H/CH.

To delineate the key amino acid residues responsible for the observed phenotypic alterations, we initially generated single-site alanine substitutions within Mut2 and Mut3 through site-directed mutagenesis. Among these, only the C1126A mutant showed a pronounced reduction in infectivity, with an approximately 419-fold decrease compared with WT (S3A Fig). Structural mapping of C1126 onto the prefusion Spike model [24] suggests the presence of a disulfide bond between C1126 and C1082 (S3B Fig). Comparative analysis of C1126 and C1082 mutants revealed that both significantly reduced infectivity (S3C Fig) by impairing Spike incorporation into SC2-VLPs, without affecting the intracellular Spike level (S3D Fig). These findings suggest that the disulfide bond between C1126 and C1082 is critical for stabilizing the Spike and enabling its incorporation into viral particles.

We individually substituted residues S1147, F1148, and K1149 within Mut9 with alanine and found that only the F1148A mutation markedly reduced SC2-VLP infectivity (Fig 3A). Structural analyses of the prefusion [24] and postfusion [13] Spike conformations revealed that F1148 mediates hydrophobic interactions within the SH trimer in the prefusion conformation (Fig 3B), whereas in the postfusion state, F1148 participates in a multi-helix assembly involving the CH and the 3H helices of adjacent protomers. The phenyl side chain of F1148 is inserted into a hydrophobic cavity formed by CH residues I993, L996 and I997, along with 3H residues I742 and C749 (Fig 3C). This spatial arrangement facilitates hydrophobic interactions between F1148 and the surrounding hydrophobic microenvironment (Fig 3C). This interaction resembles a "hydrophobic lock-and-key" mechanism, which we designate as "hydrolock". The F1148 residue in PFPH-1 is conserved across betacoronaviruses (Fig 1D). Moreover, structural alignment of the CH, 3H and PFPH regions from postfusion Spikes of SARS-CoV-2, SARS-CoV, and MHV reveals remarkable structural homology of the hydrolock (S4A Fig), with the key hydrophobic amino acids within the 3H and CH domains also being conserved (S4B and S4C Fig), highlighting the functional significance of this hydrolock across the broader betacoronaviruses.

To further validate the functional role of the hydrolock, we performed a deep scanning mutagenesis (DSM) of residue F1148 by substituting it with any other available residues. The results revealed that most substitutions severely impaired SC2-VLP infectivity, with the exception of the F1148V, F1148L, F1148W, F1148M, F1148Y, and F1148A. Notably, the mutants retaining more than 50% of the infectivity signal were those with hydrophobic side chains, underscoring the importance of hydrophobic interactions at this position (Fig 3D). Western blot analysis of the secreted SC2-VLP revealed that nearly all mutants did not affect Spike incorporation into SC2-VLPs, with the exception of F1148I (Fig 3E), consistent with findings from the Mut9 study (Fig 2B and 2D). While the F1148I mutation impaired Spike incorporation into SC2-VLPs, it did not alter Spike expression within cells. Notably, the intracellular F1148I mutant displayed a higher molecular weight compared to the WT protein, although the underlying cause of this shift remains unclear. Collectively, these data indicate that the F1148-(3H/CH) hydrophobic interaction is critical for viral entry.

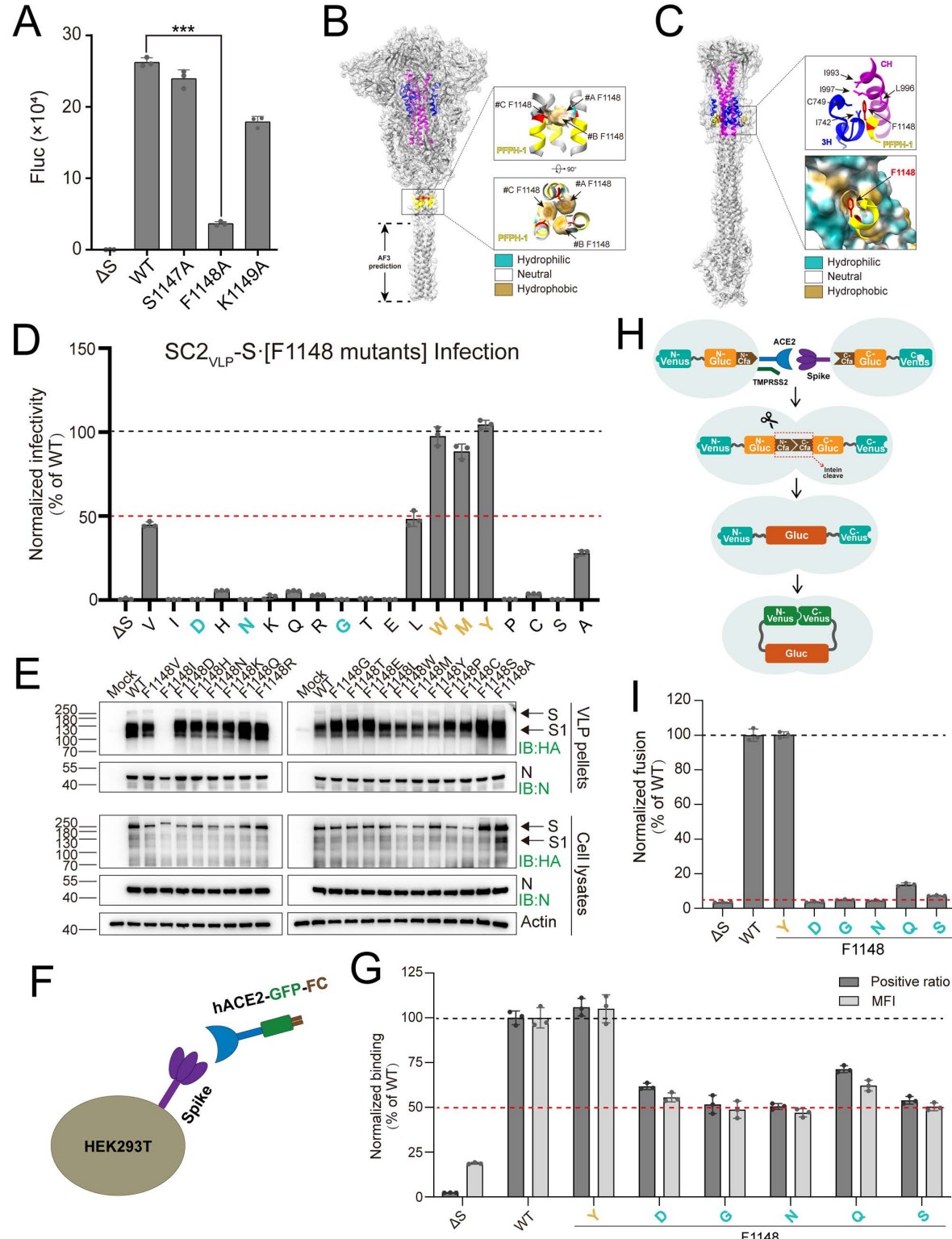

**Fig 3. Identification and characterization of a hydrophobic lock (hydrolock) interaction formed by F1148/3H/CH. (A)** The infectivity of SC2-VLP. The intracellular Firefly luciferase activities were determined at 24 hours post-infection with SC2-VLPs (mean values ± SDs, n = 3, ***P < 0.001; two-tailed, unpaired *t-t*est). (**B**) Localization of F1148 in the prefusion structure of Spike (PDB ID: 6XR8), the structure is shown in surface display mode. (**C**)

Localization of F1148 in the postfusion structure of Spike (PDB ID: 8FDW), the structure is shown in surface display mode. (**D**) The infectivity of SC2-VLP with Spike·F1148 mutants. The intracellular Firefly luciferase activities were determined at 24 hours post-infection. The relative infectivity was determined by normalizing the luciferase activity of the SC2-VLP mutants to that of WT (mean values ± SDs, n = 3). (**E**) Western blot analysis of SC2-VLPs. The SC2-VLP-containing supernatants were precipitated by methanol (Pellets) and the packaging cell lysates were subjected to Western blot assay with the indicated antibodies (IB). Mock: untransfected HEK293T control. The representative images shown for each group from multiple independent experiments. (**F**) Schematic of the Spike-ACE2 binding experimental design. Spike was expressed on the HEK293T cells surface and incubated with ACE2-GFP-Fc fusion protein solution at 4°C for 2 hours. The percentage of GFP-positive cells and the mean fluorescence intensity (MFI) was determined by flow cytometry. (**G**) Binding efficiency of Spike·F1148 mutants to ACE2. The binding efficiency was determined by normalizing the percentage of GFP-positive cells and the mean fluorescence intensity (MFI) of the Spike·F1148 mutants to that of WT (mean values ± SDs, n = 3). (**H**) Schematic of the membrane fusion experimental design. One population of HEK293T cells was co-expressed with ACE2, TMPRSS2, and a (N-Venus)-linker-(N-Gluc)-(N-intein) fusion polypeptide, and a second population of HEK293T cells was co-expressed with Spike and a (C-intein)-(C-Gluc)-linker-(C-Venus) fusion polypeptide. The two cell populations were incubated at 37°C, and the intracellular *Gaussia* luciferase activities were determined at 24 hours post-incubation. (**I**) Membrane fusion efficiency mediated by Spike·F1148 mutants. The fusion efficiency was determined by normalizing the luciferase activity of the HEK293T-Spike·F1148 mutants to that of WT (mean values ± SDs, n = 3).

## Impact of F1148 mutations on Spike-ACE2 binding and membrane fusion

To further dissect the role of the F1148-(3H/CH) hydrolock in viral entry process, we first examined how the F1148 mutation affects the Spike-ACE2 interaction, which is essential for viral entry [23]. We adapted a previously reported assay [39], in which Spike was expressed on the surface of HEK293T cells, followed by incubation with a secreted ACE2-GFP-Fc fusion protein at 4°C for 2 hours. The Spike-ACE2 binding interaction was then quantitatively assessed by flow cytometry analysis of the ACE2 receptor-positive cell populations (Fig 3F). We evaluated the F1148Y mutant that didn't affect virus entry and several mutants that severely impaired virus entry. As expected, the hydrophobic F1148Y mutant exhibited a negligible effect on Spike-ACE2 binding affinity compared to wild-type (WT). Hydrophilic mutants (F1148D, F1148G, F1148N, F1148Q, and F1148S) that severely impaired viral entry (Fig 3D) yet retained substantial ACE2-binding capacity, albeit with reduced binding kinetics (positive ratio >50%, MFI > 46% vs WT; Fig 3G). These findings suggest that the entry defect observed in virus-like particles (VLPs) bearing these hydrophilic mutants is not primarily mediated by ACE2-binding deficiency.

We then immunostained the cell-surface-expressed Spike-NTD[HA] variants using a primary anti-HA antibody followed by a FITC-conjugated secondary antibody (S5A Fig). Flow cytometric analysis showed that all hydrophilic mutations (F1148D, F1148G, F1148N, F1148Q, and F1148S) exhibited reduced yet substantial anti-HA reactivity, with positive staining ratio exceeding 60% and mean fluorescence intensity (MFI) values above 45% relative to the wild-type (WT) control (S5B Fig). Given that the hydrophilic mutations did not affect Spike incorporation into SC2-VLPs (Fig 3E), the attenuated cell surface staining is likely attributable to conformational changes in the Spike protein, which may underlie their reduced ACE2-binding capacity (Fig 3G).

Since the entry defect observed in SC2-VLPs harboring these hydrophilic mutants is not primarily attributable to impaired ACE2 binding, we next investigated the impact of these mutants on Spike-mediated membrane fusion [23]. We developed a Spike-mediated membrane fusion assay using split Cfa protein-mediated intein complementation [40]. As illustrated in Fig 3H, one population of HEK293T cells was co-transfected with ACE2, and a (N-Venus)-linker-(N-Gluc)-(N-Cfa) fusion construct, while a second population was co-transfected with Spike and a (C-Cfa)-(C-Gluc)-linker-(C-Venus) fusion construct. Upon co-incubation, Spike-mediated membrane fusion facilitated cell-cell fusion, enabling intein complementation and autocatalytic splicing to restore Gluc activity, while the flexible linker allowed Venus fluorescence reconstitution [41]. Mixing the Spike-expressing and ACE2-expressing cells resulted in robust membrane fusion, as evidenced by the strong Gluc and Venus signals (S6A and S6B Fig). Co-expression of TMPRSS2 further enhanced fusion efficiency, as evidenced by increased reporter activity (S6A and S6B Fig). Therefore, the subsequent experiments employed receptor cells co-expressing both ACE2 and TMPRSS2. Using this system, we assessed the membrane fusion activity of F1148 mutants. Consistent with its hydrophobic character, the F1148Y mutation had a minimal effect on membrane fusion,

whereas hydrophilic substitutions (F1148D, F1148G, F1148N, F1148Q, and F1148S) severely impaired membrane fusion efficiency (Fig 3I). Collectively, these data indicate that the F1148-(3H/CH) hydrophobic interaction is critical for Spike-mediated membrane fusion.

## Characterization of the 3H/CH hydrophobic lock (hydrolock) interaction

A recent study by Xing et al. [27] resolved the structure of the early fusion intermediate conformation (E-FIC) of the Spike protein, in which S1 remains attached and S2 has yet to refold. Notably, this work resolved, for the first time, a loop region spanning residues 770–831. We compared the 3H/CH hydrophobic cavities of the Spike protein across three conformational states-prefusion, E-FIC, and postfusion. Structural analysis revealed that this hydrophobic cavity remains highly conserved, with root mean square deviation (RMSD) values of 1.114 Å (prefusion vs. E-FIC), 0.554 Å (E-FIC vs. postfusion) and 1.212 Å (prefusion vs. postfusion)(Fig 4Aiv). Notably, this cavity is occupied by different hydrophobic residues in each stage: L977 in prefusion (Fig 4Ai), F782 in E-FIC (Fig 4Aii), and F1148 in postfusion (Fig 4Aiii). Structural alignment revealed that the aromatic side chains of F782 and F1148 exhibit substantial spatial overlap, indicating strong compatibility with the cavity, whereas L977 appears less optimally accommodated (Fig 4Aiv).

To elucidate the functional roles of L977 and F782 residues, which occupy the 3H/CH cavities in the prefusion and E-FIC states, respectively, we introduced hydrophilic substitutions (L977G and F782G) and assessed their impact on viral entry. Both mutations nearly completely abrogated SC2-VLP infection (Fig 4B). Mechanistic analysis revealed that the F782G substitution significantly disrupted Spike protein incorporation into SC2-VLPs, as evidenced by a marked reduction in SC2-VLP-associated Spike levels. In contrast, the L977G mutation exhibited no discernible effect on Spike assembly or SC2-VLP incorporation (Fig 4C). Structural analysis of the prefusion Spike protein revealed that F782 engages in hydrophobic interactions with I870 in the FP and V1060 in the CD; the F782G substitution likely disrupts these hydrophobic interactions, thereby destabilizing the Spike structure (S7Aiii Fig).

We further immunostained the cell-surface-expressed L977G and F782G Spike-NTD^HA mutants using the same approach as described previously (S5A Fig). The Spike·L977G mutant exhibited robust anti-HA reactivity, with a positive staining ratio of 75.0% and a mean fluorescence intensity (MFI) of 64.8%, comparable to the wild-type (WT) control. In contrast, Spike·F782G showed a markedly reduced positive staining ratio (33.5%) and MFI (33.8%) relative to WT (S8 Fig). These findings suggest that the F782G mutation, but not the L977G mutation, likely induces substantial conformational alterations in Spike folding, leading to impaired SC2-VLP incorporation.

We next focused on the L977G mutant to examine its effect on ACE-2 binding. Spike-ACE2 binding assays showed that L977G retained over 50% binding efficiency observed for WT (Fig 4D). Membrane fusion assays showed that although the L977G mutation did not affect intracellular Gluc luciferase activity (Fig 4E), Venus fluorescence imaging revealed a distinct punctate distribution pattern rather than the syncytium formation observed with WT Spike (Fig 4F). Structural analysis indicated that in the prefusion state, L977 engages the 3H/CH cavity (S7Aii Fig); during the transition to the E-FIC conformation, L977 is displaced by F782 (S7Biii Fig) while cooperating with V976 to promote zippering of the HR1 trimer (S7Bi and S7Bii Fig). This coordinated action may facilitate the ejection of HR1 and FP, with critical hydrophobic interactions persisting through the postfusion conformation until membrane fusion is complete (S7Ci and S7Cii Fig). The L977G mutation may perturb a membrane fusion step not fully captured by the co-culture assay, and its effect appears less pronounced than that of F1148G, which is sufficient to surpass the detection threshold of the cell-cell membrane fusion system. These findings suggest that L977 plays a critical role in the fusion process.

Given the structural conservation of the 3H/CH region across multiple conformational stages, we propose that 3H-CH interaction is highly stable. Structural analysis revealed that hydrophobic interactions are formed between 3H residues I742, L752, L753, L756, L759, and L763 and CH residues I993, I997, L1001, and L1004, which likely stabilize the 3H/CH interface (Fig 4G). Notably, some residues, such as I742 (3H) and I993/I997 (CH), contribute both to 3H-CH interactions and the assembly of the 3H/CH cavity (Fig 4G). To determine whether key hydrophobic residues within the 3H/CH cavity

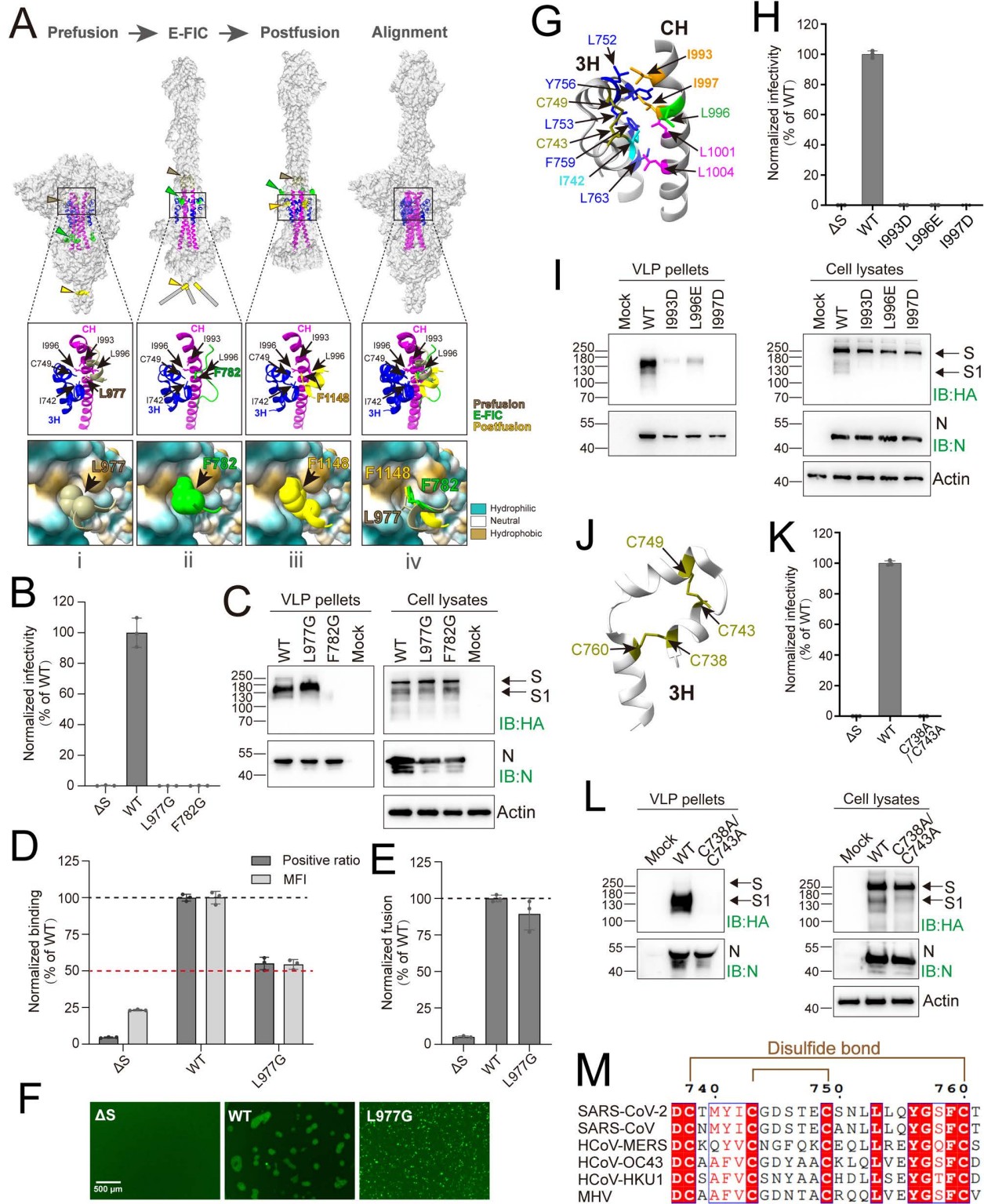

**Fig 4. Characterization of the 3H/CH hydrophobic lock (hydrolock) interaction. (A)** Schematic of the "hydrolock interaction" model. i, prefusion (PDB ID: 6XR8); ii, early fusion intermediate conformation (E-FIC) (PDB ID: 8Z7P). iii, postfusion (PDB ID: 8FDW). **(B)** The infectivity of SC2-VLPs. Intracellular Firefly luciferase activities were determined at 24 hours post-infection (mean values ± SDs, n = 3). **(C)** Western blotting of SC2-VLPs. The

SC2-VLP-containing supernatants precipitated by methanol (Pellets) and the packaging cell lysates were subjected to Western blotting assay with the indicated antibodies (IB). Mock: untransfected HEK293T control. The representative images shown for each group from multiple independent experiments. **(D)** Binding efficiency of Spike to ACE2. The binding efficiency was determined by normalizing the percentage of GFP-positive cells and the mean fluorescence intensity (MFI) of the Spike mutants to that of WT (mean values ± SDs, n = 3). **(E)** Membrane fusion efficiency mediated by Spike·L977G. The fusion efficiency was determined by normalizing the luciferase activity of the HEK293T-Spike·F1148 mutants to that of WT (mean values ± SDs, n = 3). **(F)** Membrane fusion efficiency mediated by Spike. The fluorescence images were acquired at 24 hours post-incubation. **(G)** Structural visualization of the interaction sites between 3H and CH in the postfusion structure of Spike (PDB ID: 8FDW). **(H)** The infectivity of SC2-VLPs. Intracellular Firefly luciferase activities were determined at 24 hours post-infection (mean values ± SDs, n = 3). **(I)** Western blotting of SC2-VLPs. The SC2-VLP-containing supernatants precipitated by methanol (Pellets) and the packaging cell lysates were subjected to Western blotting assay with the indicated antibodies (IB). Mock: untransfected HEK293T control..The representative images shown for each group from multiple independent experiments. **(J)** Structural visualization of the disulfide bonds in the 3H in the postfusion structure of Spike (PDB ID: 8FDW). **(K)** The infectivity of SC2-VLPs. Intracellular Firefly luciferase activities were determined at 24 hours post-infection (mean values ± SDs, n = 3). **(L)** Western blotting of SC2-VLPs. The SC2-VLP-containing supernatants precipitated by methanol (Pellets) and the packaging cell lysates were subjected to Western blotting assay with the indicated antibodies (IB). Mock: untransfected HEK293T control. The representative images shown for each group from multiple independent experiments. **(M)** Alignment of the betacoronavirus Spike 3H domains was performed using Clustal Omega and ESPript 3.0. Red highlighting represents 100% identity, whereas blue boxed red fonts shows a global score of 70% identity based on ESPript 3.0 parameters.

functionally mimic their conformation-specific partners (L977, F782 and F1148), we introduced hydrophilic substitutions in CH (I993D, L996E, and I997D). The infectivity of SC2-VLP-(S·I993D), SC2-VLP-(S·L996E), and SC2-VLP-(S·I997D) was nearly abolished (Fig 4H), and Western blot analysis demonstrated that these mutations severely impaired Spike incorporation into SC2-VLPs while leaving intracellular expression unaffected (Fig 4I, IB: HA). Among these, I997D almost entirely abrogated Spike assembly on SC2-VLPs (Fig 4I, IB: HA).

Further structural analysis revealed that 3H contains two disulfide bonds, C738-C760 and C743-C749, which stabilize its tertiary structure and promote 3H interaction (Fig 4J). The C743-C749 bond, in particular, is directly involved in forming the 3H/CH cavity (Fig 4J). These four cysteine residues are fully conserved across the betacoronaviruses (Fig 4M). Disruption of both disulfide bonds via a double mutation C738A/C743A severely impaired the SC2-VLP infectivity (Fig 4K) and almost abolished Spike assembly on the SC2-VLPs without affecting intracellular expression (Fig 4L). Collectively, these findings highlight that while direct pairwise interactions between the key hydrophobic residues (L977, F782, and F1148) and the 3H/CH cavity residues remain unconfirmed, the hydrophobic core of the 3H/CH region, together with its disulfide bond-stabilized structural integrity, is functionally critical for proper Spike assembly on SC2-VLPs and, by extension, viral infectivity.

### Structural analysis of anti-SH mAbs that target PFPH-1 and 3H/CH interactions

Several neutralizing antibodies targeting the stem region have been identified, most of which recognize epitopes within residues 1142–1167, with F1148 in the PFPH-1 motif serving as a critical determinant for antibody binding [31–35,42–44]. To further elucidate these interactions, we performed structural alignments of the resolved antibody-epitope complexes with the postfusion Spike protein. The epitope peptides recognized by antibodies S2P6 [31], CC40.8 [33], WS6 [35] and CV3–25 [45] exhibited root mean square deviation (RMSD) values of 0.996 Å, 1.127 Å, 0.918 Å, and 0.710 Å, respectively, relative to the postfusion Spike structure (Fig 5A). The results revealed substantial steric clashes between these antibodies and the postfusion Spike, suggesting that these antibodies likely engage their epitopes during SH refolding, prior to the stabilization of the fully postfusion conformation [26]. We next performed structural alignment of these neutralizing antibodies with the hydrolock PFPH-1/3H/CH complex. The analysis revealed that S2P6, CC40.8, and WS6 specifically target the PFPH-1 domain and exhibit spatial clashes with the 3H/CH region. In contrast, CV3–25, which binds to the C-terminal region adjacent to PFPH-1, does not directly clash with 3H/CH but instead interferes spatially with HR1, located near the N-terminus of CH (Fig 5B). These findings suggest that neutralization can occur not only through direct engagement of residues within the hydrolock but also by targeting adjacent structural elements, either via direct blockade of hydrolock interactions or through steric hindrance that disrupts its functional integrity.

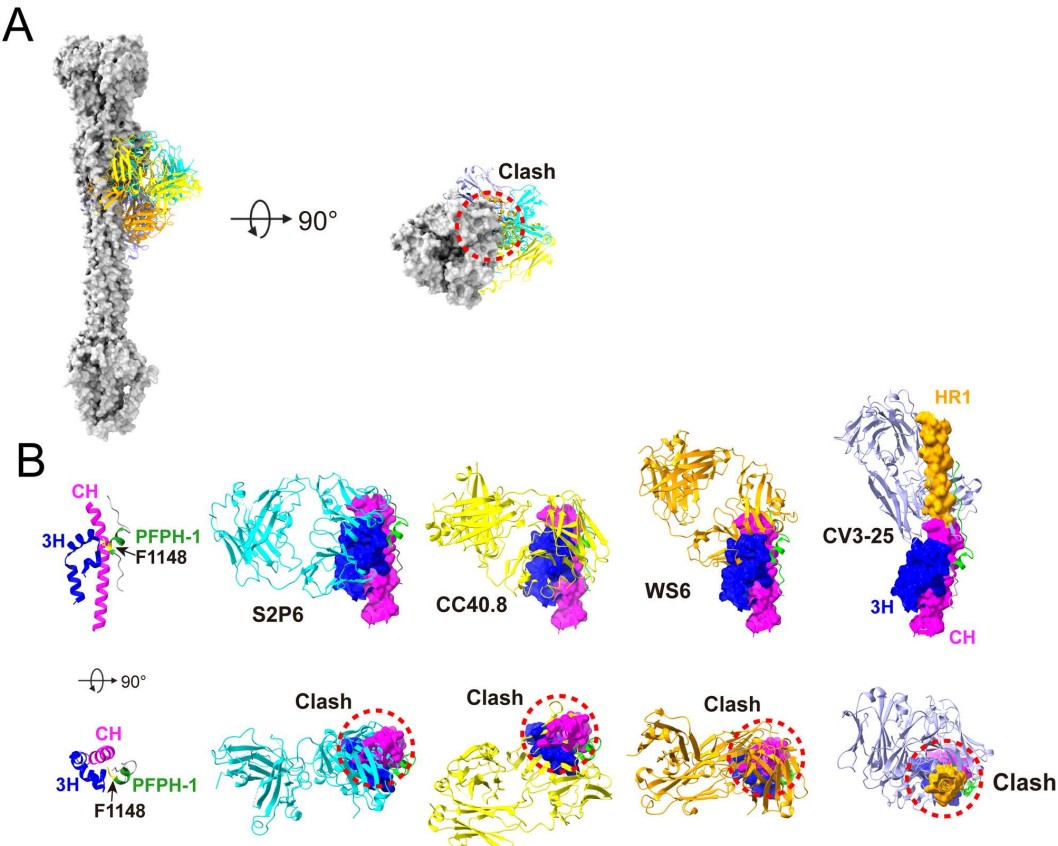

**Fig 5. Anti-SH mAbs disrupt PFPH-1 and 3H/CH Interactions. (A)** The structural alignment of postfusion Spike with anti-SH mAbs. The structural alignment was generated by aligning postfusion Spike (PDB ID: 8FDW) with S2P6 (PDB ID: 7RNJ), CC40.8 (PDB ID: 7SJS), WS6 (PDB ID: 7TCQ) and CV3-25 (PDB ID: 7RAQ) using ChimeraX software. **(B)** Enlarged region of each anti-SH mAb in panel **A**. The first column shows structure of PFPH-1 and 3H/CH, with the F1148 residue in PFPH-1 is indicated in yellow. The second, third, fourth, and fifth columns depict the structural alignment of S2P6 (cyan), CC40.8 (yellow), WS6 (orange), and CV3-25 (lavender) with PFPH-1, respectively.

## Evaluation of neutralization capacity directed at regions proximal to PFPH-1

To further validate the neutralization potential of targeting the hydrolock interaction, we utilized a replacement mutagenesis strategy in which each 9-residue segment of the Spike SH region was substituted with an exogenous linear epitope HA (YPYDVPDYA). The neutralization sensitivity of each SH segment was evaluated by measuring SC2-VLPs infectivity following incubation with an anti-HA monoclonal antibody (mAb) (Fig 6A). Replacement of residues within PFPH-1 (SC2-VLP-(1142-S-SH$^{HA}$) and SC2-VLP-(1151-S-SH$^{HA}$) and PFPH-3 (SC2-VLP-(1178-S-SH$^{HA}$) and SC2-VLP-(1187-S-SH$^{HA}$) resulted in a marked reduction in infectivity (Figs 1C and S9A), consistent with the findings from the alanine scanning mutagenesis (Fig 2B). Substitution in other SH regions retained sufficient infectivity for subsequent neutralization analysis.

Western blotting confirmed that Spike incorporation into SC2-VLPs was only slightly affected by 1133-S-SH$^{HA}$ and 1160-S-SH$^{HA}$ mutants, while other substitutions had no impact, and all mutants exhibited comparable intracellular Spike expression (S9B Fig). Neutralization assays with varying concentrations of anti-HA mAb demonstrated inhibition of infection of SC2-VLP infectivity for HA insertions at residues 1133–1141, 1160–1168, and 1196–1204, with IC$_{50}$ values of 3569 ng/mL, 44.99 ng/mL, and 324.90 ng/mL, respectively (Fig 6B), whereas the IgG control showed no neutralizing activity (S10 Fig).

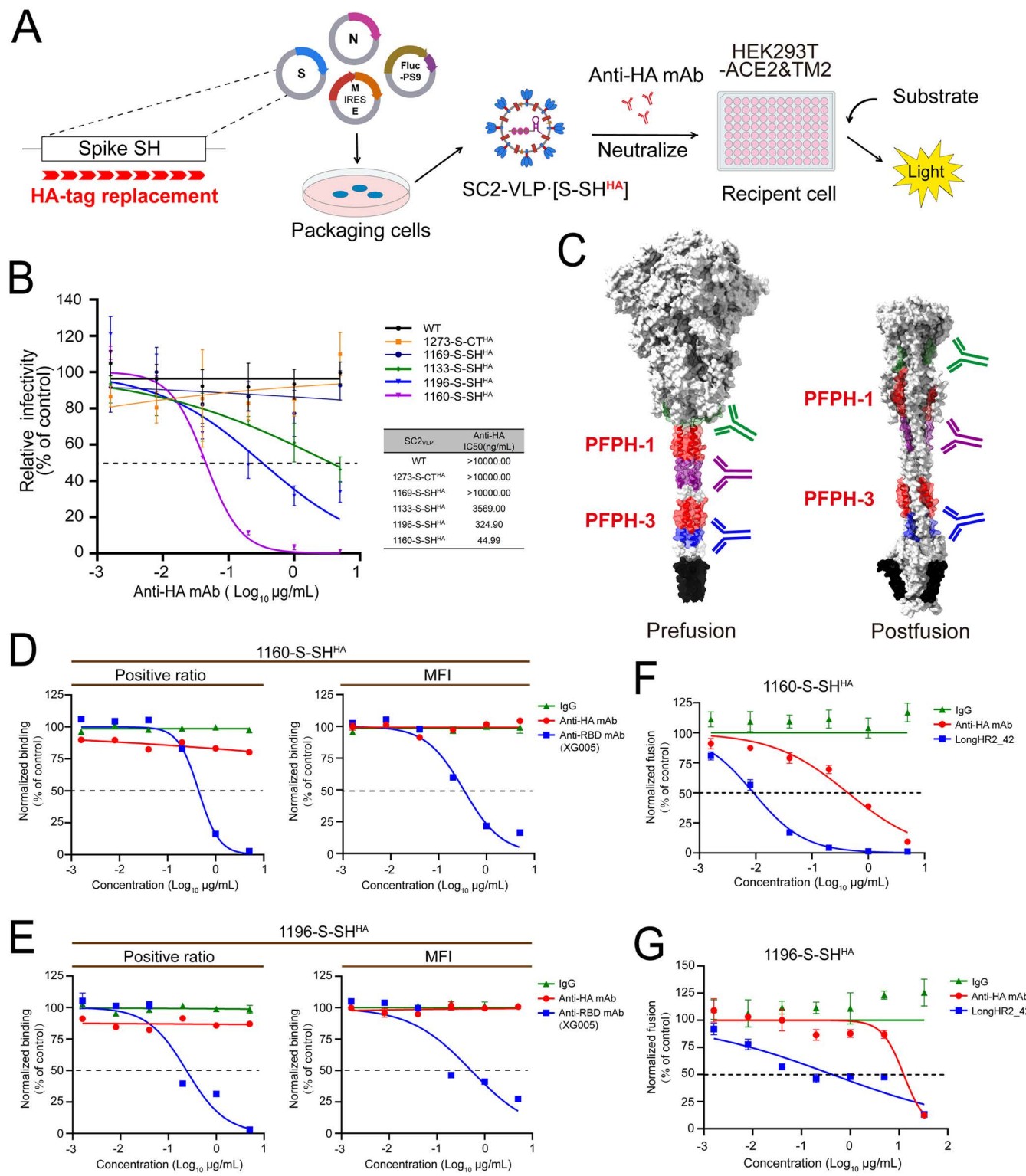

**Fig 6. Neutralization potential of the regions adjacent to PFPH-1 and PFPH-3. (A)** Schematic of the neutralization experimental design. The Spike SH region was replaced with an HA-tag to generate SC2-VLP-(S-SH^HA), which were incubated with anti-HA monoclonal antibody at indicated concentration at 37°C for 1 hour, followed by infection of HEK293T-ACE2&TMPRSS2 cells. The intracellular Firefly luciferase activities were determined at 24

hours post-infection. **(B)** Dose-dependent neutralization of SC2-VLP-(S-SH<sup>HA</sup>) infectivity by anti-HA mAb (5-fold serial dilution). The intracellular Firefly luciferase activities were determined at 24 hours post-infection. The relative infectivity was determined by normalizing the luciferase activity of SC2-VLPs incubated with antibodies to antibody-free controls (mean values ± SDs, n = 3). **(C)** Visualization of the regions with effective neutralization activity in the prefusion (PDB ID: 6XR8 and AlphaFold3-predicted SH trimer) and postfusion (PDB ID: 8FDW) conformations of the Spike protein. Key neutralizing regions are color-coded: residues 1133-1141 (green), residues 1160-1168 (purple), and residues 1196-1204 (blue), along with PFPH-1 (red), PFPH-3 (red), are indicated. **(D-E)** Binding efficiency of Spike·SH<sup>HA</sup> mutants to ACE2 (Panel D and E are 1160-S-SH<sup>HA</sup> and 1196-S-SH<sup>HA</sup>, respectively). The binding efficiency was determined by normalizing the percentage of the mean fluorescence intensity (MFI) and GFP-positive HEK293T-Spike·SH<sup>HA</sup> cells incubated with anti-HA mAb, IgG (Negative control) and XG005 (Positive control) to that of HEK293T-Spike·SH<sup>HA</sup> incubated with medium as a control (mean values ± SDs, n = 3). **(F-G)** Membrane fusion efficiency mediated by Spike·SH<sup>HA</sup> mutants (Panel F and G are 1160-S-SH<sup>HA</sup> and 1196-S-SH<sup>HA</sup>, respectively). The fusion efficiency was determined by normalizing the luciferase activity of the HEK293T-Spike·SH<sup>HA</sup> incubated with anti-HA mAb, IgG (Negative control) and LongHR2_42 peptide (Positive control) to that of HEK293T-Spike·SH<sup>HA</sup> incubated with medium as a control (mean values ± SDs, n = 3).

Structural mapping revealed that these neutralization-sensitive regions are adjacent to PFPH-1 and PFPH-3, with the strongest neutralization observed for the 1160–1168 region near PFPH-1 (Fig 6C). Collectively, these results suggest that antibodies targeting PFPH-1 and PFPH-3, or their adjacent regions—particularly those near PFPH-1—exert effective neutralizing activity.

## Inhibition of membrane fusion through targeting regions proximal to PFPH-1

To elucidate the mechanistic basis of antibody-mediated neutralization targeting regions adjacent to PFPH-1, we initially examined the correlation between antibody accessibility and neutralization efficacy. To this end, we developed a sandwich ELISA assay to assess the accessibility of HA epitopes in SC2-VLP-(S-SH<sup>HA</sup>) with an anti-HA mAb. SC2-VLP-(S-SH<sup>HA</sup>)-containing supernatants were incubated on plates pre-coated with anti-RBD antibodies, which captured the particles, and bound SC2-VLP-(S-SH<sup>HA</sup>) was subsequently detected with an anti-HA primary antibody and an HRP-conjugated secondary antibody (S11A Fig). Compared with 23-S-NTD<sup>HA</sup> (HA insertion in NTD) and M<sup>HA</sup> (HA insertion in the N-terminus of M), SC2-VLP-1133-S-SH<sup>HA</sup>, SC2-VLP-1160-S-SH<sup>HA</sup> and SC2-VLP-1196-S-SH<sup>HA</sup>, which contain HA replacements adjacent to PFPH-1 and PFPH-3 within SH, exhibited reduced accessibility to the anti-HA mAb (S11B Fig), despite being effectively neutralized by the same anti-HA mAb (Fig 6B). This finding aligns with a recent molecular dynamics (MD) simulation study showing that the antibody-accessible surface area (AbASA) of SH increases during early S2 refolding, as opposed to its prefusion or postfusion conformations [26]. The lack of the S2 hairpin intermediate in the SC2-VLPs used for ELISA likely accounts for the reduced antibody accessibility to SH.

To investigate which step of the viral entry is blocked by the anti-HA mAb targeting regions adjacent to PFPH-1 and PFPH-3, we evaluated both ACE2 binding and membrane fusion mediated by the Spike.1160-S-SH<sup>HA</sup> and Spike.1196-S-SH<sup>HA</sup>, using the previously described Spike-ACE2 binding assay and Spike-mediated membrane fusion assay. In the binding assay, XG005 [46], an anti-RBD nAb, inhibited ACE2 binding in a dose-dependent manner, whereas neither the anti-HA mAb nor the IgG control affected (Spike-SH<sup>HA</sup>)-ACE2 interactions (6D and 6E Fig). In contrast, in the membrane fusion assay, the anti-HA mAb, similar to LongHR2_42 [47] —a membrane fusion-inhibitory HR1 peptide—dose-dependently inhibited (Spike-SH<sup>HA</sup>)-mediated membrane fusion (Fig 6F and 6G). These findings indicate that the anti-HA mAb neutralizes viral entry by specifically inhibiting membrane fusion rather than interfering with Spike-ACE2 binding.

## Discussion

The SH region of coronavirus Spike protein plays a crucial role in membrane fusion and serves as a conserved target for broad-spectrum antiviral strategies. However, the detailed mechanism by which SH mediates S2 subunit refolding remains incompletely understood. The HR1/HR2 6-HB is widely recognized as the core structure of membrane fusion [29,30]. Here, we structurally defined three postfusion-preserved helices (PFPHs) that retain α-helices conformations in both

prefusion and postfusion states of Spike. Of which, the PFPH-3 interacts with HR1 to form the 6-HB (Figs 1C and 2C). Functional analysis revealed that alanine substitutions within PFPH-1 caused a ~ 122-fold reduction in SC2-VLP infectivity, whereas substitutions within PFPH-3 led to a more modest ~16-fold reduction (Fig 2B), indicating that PFPH-1 plays a more critical role in virus entry.

Mutagenesis analysis identified F1148 within PFPH-1 as a critical determinant of SC2-VLP infectivity. Deep mutational scanning and functional studies demonstrated that only hydrophobic residues at position F1148 were tolerated (Fig 3D). In contrast, Western blot detection of the HA-tag inserted into the NTD domain revealed that hydrophilic substitutions at this site did not impair Spike incorporation into SC2-VLPs (Fig 3E). However, the limitation lay in the abolished affinity of anti-S2 antibody for recognizing Spike·F1148 mutants (S12 Fig), which prevented us from further determining the specificity of Spike incorporation into SC2-VLPs. Flow cytometric analysis of cell-surface-expressed hydrophilic NTD^HA mutants showed reduced yet substantial anti-HA staining (S5B Fig), which may contribute to their attenuated ACE2 binding (Fig 3G). These hydrophilic substitutions likely induce conformational changes in the Spike protein, thereby affecting NTD reactivity with antibody. In the prefusion Spike, the F1148 mediates hydrophobic interactions within the SH trimer (Fig 3B) and hydrophilic substitutions that disrupt these interactions may partially compromise overall spike folding. Notably, despite retaining substantial ACE2-binding activity, all hydrophilic substitutions at F1148 almost completely abolished membrane fusion (Fig 3I), demonstrating that F1148 is essential for the fusion process rather than ACE2 binding.

Structurally, in the postfusion conformation, F1148 is tightly embedded within a hydrophobic cavity formed by 3H/CH, generating strong hydrophobic interactions (Fig 3C). We hypothesize that this hydrophobic "lock-and-key" interaction, which we term the 'hydrolock', serves as a driving force for refolding of the prehairpin intermediate during membrane fusion. Comparative structural analyses of various conformational states revealed that the 3H/CH cavity is highly conserved, being sequentially occupied by hydrophobic residues: L977 in the prefusion state, F782 in the early fusion intermediate conformation (E-FIC), and F1148 in the postfusion state (Fig 4A). Structural alignment further revealed that the aromatic side chains of F782 and F1148 optimally fit the cavity, while L977 is less ideally accommodated (Fig 4Aiv). This suggests that substitution of L977 with F782 is energetically favorable, whereas replacement of F782 by F1148 is a more energetically demanding step, underscoring the pivotal role of the F1148-3H/CH interaction in driving refolding.

Based on these findings, we propose a model in which PFPH-1 (proximal to the Spike head) and PFPH-3 (proximal to the membrane) orchestrate S2 refolding (Fig 7A). Following ACE2 engagement and S1 shedding (Fig 7Ai and 7Aii), the fusion peptide (FP) of the S2 subunit protrudes and anchors to the host cell membrane. The F1148 position in PFPH-1 does not affect FP insertion, as SH remains largely unfolded in the E-FIC state while FP is already extended [27] (Fig 4Aii). We propose that F1148 engages the 3H/CH hydrophobic cavity early during S2 refolding, acting as a molecular trigger for ensuing conformational transitions (Fig 7Aiii). In the middle stage, PFPH-3, anchored by F1148, facilitates zipping of the refolding helices (Fig 7Aiv and 7Av). In the late stage, PFPH-3 engages with HR1 to form the 6-HB, completing membrane fusion (Fig 7Avi). A previous all-atom molecular dynamics simulation study suggested that glycan shielding may slow S2 refolding and preferentially hinders PFPH-3 refolding, leaving PFPH-1 unaffected [28], which may allow early engagement of F1148 with the 3H/CH cavity to initiate S2 refolding (Fig 7Aiii).

Based on our model, antibodies targeting PFPH-1 and its proximal regions impede the early stages of S2 refolding (Fig 7B and 7Ci), whereas antibodies directed toward the vicinity of PFPH-3 interfere with late refolding steps, thereby inhibiting membrane fusion (Fig 7Cii). Most reported broadly neutralizing antibodies (bnAbs) targeting the SH region recognize PFPH-1, with F1148 identified as a key binding residue [14]. HA-tag insertion and neutralization assays revealed that adjacent regions of PFPH-1, including residues 1133–1141 and 1160–1168, also mediate neutralizing activity by interfering with membrane fusion (Fig 6B, 6C, 6F, and 6G). Notably, the neutralizing efficacy of PFPH-1-targeting antibodies was not correlated with antibody accessibility (S11B Fig), suggesting that their inhibitory effect arises from disrupting the hydrophobic interaction between F1148 and the 3H/CH cavity during prehairpin refolding. Antibodies targeting the region

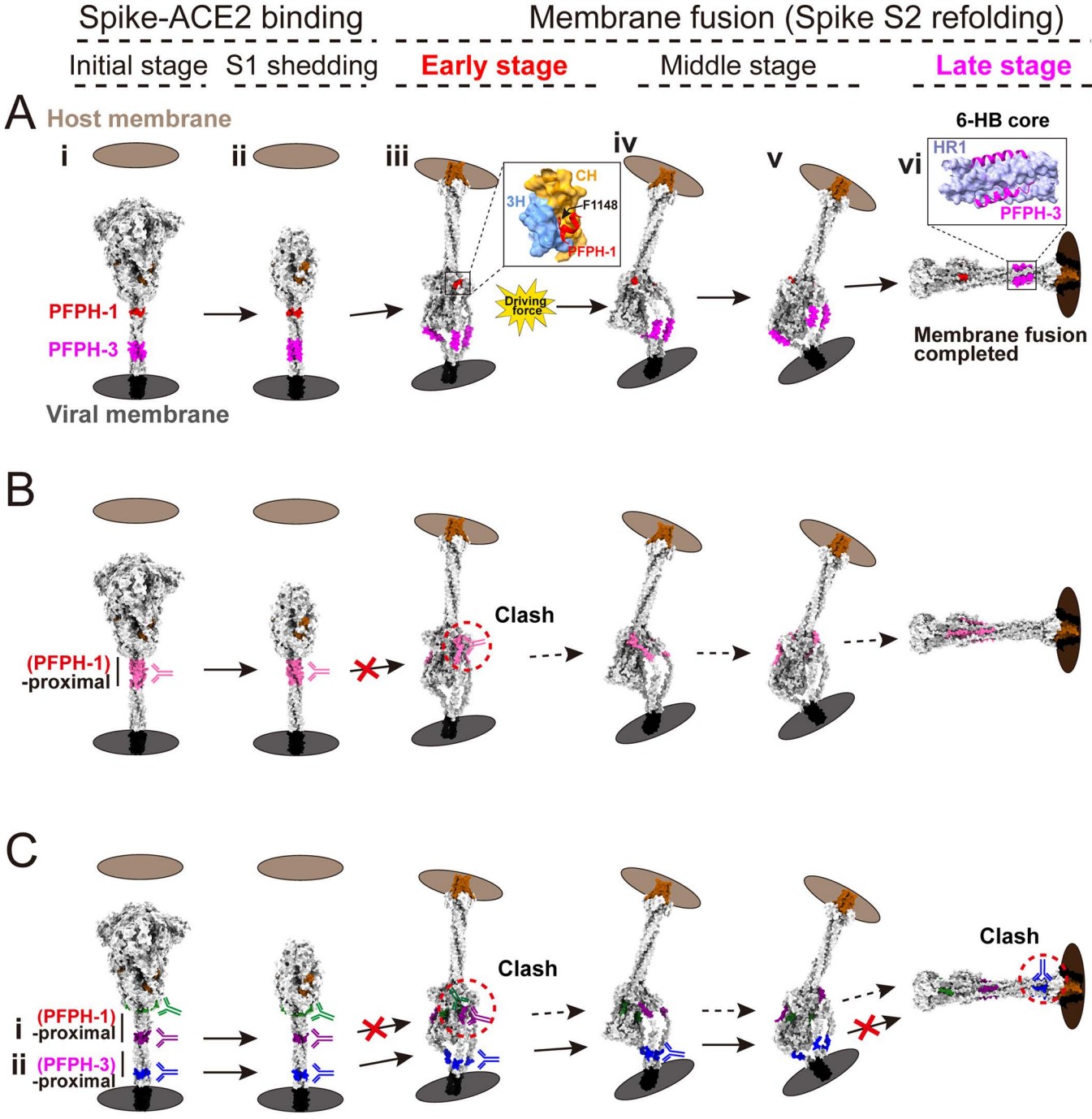

**Fig 7. Mechanistic models of S2 refolding and antibody-mediated neutralization. (A)** Mechanistic model of the key domains and residues involved in the S2 refolding mechanism in the membrane. The image was generated using the simulated intermediate atomic models from Dodero-Rojas et al. [28], the simulated intermediate MD models from Wenwei Li et al. [26] and PDBs 6XR8 and 8FDW. The PFPH-1 (red), PFPH-3 (magenta), cavity assembly by 3H (light blue) and CH (orange), and six-helix bundle (6-HB) assembly by PFPH-3 and HR1 (lavender), are indicated. **(B-C)** Spatial relationships between neutralizing epitopes and structural domains. Reported antibody-targeted epitope regions near the PFPH-1 (panel B, residues 1142-1167, hot pink), as well as the effective neutralizing activity regions near the PFPH-1 (panel C i, residues 1133-1141, green; residues 1160-1168, purple) and near the PFPH-3 (panel C ii, residues 1196-1204, blue) identified in this study, are shown.

near PFPH-3 also exhibited neutralizing activity (Fig 6B and 6C), likely by interfering with late-stage prehairpin refolding and 6-HB formation. An elegant study by Li et al. [26] captured the prehairpin intermediate structure of the Spike protein demonstrated that the antibody CV3–25 binds to residues 1149–1167 within this intermediate. These findings collectively indicate a shared mechanism among antibodies targeting the F1148 hydrolock to prevent membrane fusion. Thus, the F1148-mediated hydrolock interaction identified in this study represents a novel target for the development of broad-spectrum antiviral strategies.

## Materials and methods

### Cell culture

The human embryonic kidney cell line HEK293T was purchased from the Cell Bank of the Chinese Academy of Sciences (Shanghai, China, www.cellbank.org.cn) and maintained in Dulbecco's modified medium supplemented with 10% FBS (Gibco), 1% penicillin/streptomycin (Biological Industries) and 25 mM HEPES (Gibco).

### Plasmids

All SARS-CoV-2 sequences used in this study correspond to the nCoV-SH01 strain (accession number MT121215). Codon-optimized genes encoding the SARS-CoV-2 structural protein S, M, and E were synthesized by BGI (Wuxi, China). The S gene fragment was cloned into the KpnI/NotI sites of the phCMV vector (Genlantis) to generate phCMV-S. To construct pTRIP-IRES-M-E, the M gene fragment was first inserted into the XhoI/BamHI sites of the pTRIP vector to create pTRIP-M, followed by insertion of the EMCV IRES and E gene fragments, assembled by fusion PCR, into the BamHI/MluI using a Hieff Clone Plus One Step Cloning Kit (Yeasen). The N gene fragment was cloned into the KpnI/NotI sites of phCMV to generate phCMV-N plasmid, and the S202R mutation was introduced by fusion PCR to produce phCMV-N·S202R. For luciferase reporter construction, the firefly luciferase gene was first cloned into the HindIII/EcoRV sites of the pCDNA3.1 (Invitrogen) to generate pFluc, and the packaging sequence PS9 [36], amplified from the subgenomic replicon plasmid pBAC-sgnCoV-sGluc [48], was inserted into the EcoRV/XhoI sites of pFluc to yield pFluc-PS9. An HA-tag (YPYDVPDYA) was inserted immediately after the first residue of M via fusion PCR to create pTRIP-IRES-M·(1-NT$^{HA}$)-E. HA-tag insertions or replacement in the S gene fragment were also performed by using fusion PCR, generating a series of phCMV-S·HA plasmids. Deep mutational scans of F1148, along with alanine mutations, were introduced into phCMV-S·(23-NTD$^{HA}$) via fusion PCR, yielding phCMV-S·(23-NTD$^{HA}$)·Mut constructs. The detailed mutation information provided in S1 Table. To generate pTRIP-IRES-Puro-ACE2-myc, human ACE2 cDNA (NM-021804) was synthesized (BGI, Suzhou) and cloned into the XbaI/BsrGI sites of pTRIP-IRES-Puro backbone [49]. Plasmid pV1-BSD-TM2 was generated by amplifying human TMPRSS2 (kindly gifted by Zhang Rong, Fudan University) and cloning it into the pV1-BSD backbone after SfiI digestion. Plasmid phCMV-ACE2-GFP-Fc was constructed following a strategy reported previously with modification [39], where a SLAM signal peptide (MDPKG LLSLT FVLFL SLAFG) was fused to the N-terminus of ACE2 ectodomain (residues 20–740), and a cassette containing superfolder GFP (sfGFP) and an Fc fragment of human IgG (sfGFP-Fc) were fused to its C-terminus. The sequences of ACE2-GFP-Fc were provided in the Supporting information. For the split-intein fusion system, pTRIP-IRES-BSD-VNnGluc-cfaN was generated by assembling codon-optimized sequences encoding the N-terminal portion of Venus (VN, residues 1–172), the N-terminal portion of *Gaussia* luciferase (nGluc, residues 1–93), and the N-terminal portion of the Cfa intein (CfaN, residues 1–101) [40], which were cloned into the XbaI/BamHI sites of pTRIP-IRES-puro. Similarly, pTRIP-IRES-puro-CfaC-cGlucVC was constructed by assembling codon-optimized sequences encoding the C-terminal portion of the Cfa intein (CfaC, residues 102–136) [40], the C-terminal portion of Gluc (cGluc, residues 94–169), and the C-terminal portion of Venus (VC, residues 173–239), were cloned into the same sites. The sequences of VNnGluc-cfaN and CfaC-cGlucVC were provided in the Supporting information. All plasmid constructs were confirmed by Sanger sequencing. Detailed plasmid information is available upon request.

## SC2-VLP production

HEK293T cells ($1.125 \times 10^6$) were seeded onto poly-L-lysine-coated 6-well plates. After 24 hours, cells were transfected with a total of 4 μg plasmid DNA consisting of 0.0192 μg S, 1.96 μg M-E, 0.92 μg N·S202R, and 1.12 μg Fluc-PS9 using PEI transfection reagent (Yeasen, 40816ES01), following the manufacturer's protocol. The culture medium was replaced 8 hours post-transfection. At 72 hours post-transfection, the supernatant containing SC2-VLPs was collected, filtered through a 0.45-μm membrane (Millex), supplemented with 25 mM HEPES (Gibco), and stored at -80°C.

## Antibodies and peptide

Anti-HA rabbit monoclonal antibody (Cell Signaling Technology, #3724) was used for Western blotting at a 1:2000 dilution. Anti-SARS-CoV-2 Spike RBD rabbit monoclonal antibody (GeneTex, GTX635692) and anti-SARS-CoV-2 Spike S2 mouse monoclonal antibody (GeneTex, GTX632604) were used at 1:1000 dilution. Anti-SARS-CoV-2 Nucleocapsid rabbit monoclonal antibody (GeneTex, GTX635679), anti-SARS-CoV-2 Membrane rabbit monoclonal antibody (GeneTex, GTX636245), and anti-SARS-CoV-2 Envelope rabbit monoclonal antibody (GeneTex, GTX136046) were used at 1:2000 dilution. Anti-β-Actin antibody (Sigma; A1978) was used at 1:2000 dilution as a loading control. Human IgG (Yeasen, 36110ES03) and Rabbit IgG (Yeasen, 36113ES10) were used as negative controls for the XG005 nAb and anti-HA mAb, respectively, with dilutions adjusted to experimental requirements. The neutralizing antibody XG005 (anti-RBD nAb), kindly gifted by prof. Qiao Wang (Fudan University, China) and described previously [46], was serially diluted as specified in the experimental protocol. The peptide LongHR2_42 (PDVDL GDISG INASV VNIQK EIDRL NEVAK NLNES LIDLQ EL), designed as reported in [47], was synthesized by GL Biochem (Shanghai) Ltd. And serially diluted as required.

## Western blotting

For analysis of SC2-VLP pellets, 360 μL of SC2-VLP-containing supernatant was mixed with four volumes of methanol and incubated at room temperature for 5 minutes. The mixture was centrifuged at 12,000 rpm for 15 minutes at 4°C, and the resulting pellet was air-dried to remove residual methanol, then resuspended in 40 μL 2 × SDS loading buffer (100 mM Tris-Cl [pH 6.8], 4% SDS, 0.2% bromophenol blue, 20% glycerol, 10% 2-mercaptoethanol) and boiled for 15 minutes. For cell lysates, the cells cultured in 6-well plate were washed with PBS and then lysed with 600 μL of 2 × SDS loading buffer per well, followed by boiling for 15 minutes. For M protein detection, the boiling step was omitted, and the cell lysates were subjected to 20 passes through a 1 mL syringe needle to shear genomic DNA. Subsequently, 10 μL of each sample was loaded onto an SDS-PAGE gel, and separated proteins were transferred to a nitrocellulose membrane (Cytiva). The membrane was blocked with PBS containing 5% milk and 0.05% Tween-20 for 1 hour, then incubated with primary antibodies diluted in blocking buffer. After three washes with PBST (PBS, 0.05% Tween-20), the membranes were incubated with secondary antibodies. Following three additional washes with PBST, protein detection was performed using the Western Lightning Super ECL Detection Reagent (Yeasen, cat. no. 36208ES).

## SC2-VLPs infection

SC2-VLPs infection was carried out as previously described [36]. To generate receiver cells overexpressing ACE2 and TMPRSS2, HEK293T cells ($1.125 \times 10^6$) were seeded onto poly-L-lysine-coated 6-well plates. On the following day, cells were transfected with 4 μg of total plasmid DNA at a 1:1 ratio using the Lipofectamine 3000 transfection kit (Invitrogen, L3000015) according to the manufacturer's protocol. The medium was refreshed 8 hours post-transfection. At 24 hours post-transfection, the cells were harvested by trypsinization, resuspended, and mixed with 50 μL of SC2-VLPs-containing supernatant. The cell-virus mixture was seeded into 96-well plates at a density of $1.5 \times 10^4$ cells per well. After 24 hours of infection, the supernatant was removed, and the cells were washed with PBS before adding 50 μL of 1 × passive lysis buffer (Promega). Subsequently, 30 μL of the lysate was mixed with 50 μL of Firefly luciferase substrate (Promega), and luminescence was measured using a GLOMX luminometer (Promega).

## Neutralization assay

Anti-HA rabbit mAb and rabbit IgG control were diluted (5-fold), and 2.22 µL of each diluted antibody solution was mixed with 20 µL of SC2-VLP-containing supernatant. The mixture was incubated at 37°C for 1 hour to allow antibody-virus interaction. HEK293T cells transiently co-transfected with ACE2 and TMPRSS2 were harvested by trypsin digestion, resuspended, and added to the antibody-incubated SC2-VLP mixture. The cells were then seeded onto 96-well plates at a density of $1.5 \times 10^4$ cells per well. Luciferase activity, reflecting SC2-VLP infectivity, was measured 24 hours post-infection. Dose-response curves were generated and fitted using GraphPad Prism software to calculate the half-maximal inhibitory concentration ($IC_{50}$) for each antibody.

## SC2-VLP immunoprecipitation

For SC2-VLP immunoprecipitation, 720 µL of freshly harvested SC2-VLP-containing supernatant was divided into two equal aliquots of 360 µL each. One aliquot was subjected to methanol precipitation, and the resulting pellet was resuspended in 40 µL of 2×SDS loading buffer (100 mM Tris-Cl [pH 6.8], 4% SDS, 0.2% bromophenol blue, 20% glycerol, 10% 2-mercaptoethanol) to serve as the input control. The another aliquot was incubated with 10 µL of anti-HA magnetic beads (Thermo, 88837) at room temperature with gentle rotation for 2 hours, followed by three washes with culture medium. The beads were subsequently resuspended in 40 µL of 2×SDS loading buffer and boiled for 15 minutes prior to analysis.

## Spike-ACE2 binding assay

To produce soluble ACE2-GFP-Fc fusion protein, HEK293T cells ($1.125 \times 10^6$) were seeded onto poly-L-lysine-coated 6-well plates. On the following day, cells were transfected with 4 µg of ACE2-GFP-Fc plasmid using PEI transfection reagent (Yeasen, 40816ES01) according to the manufacturer's instructions. The medium was refreshed 8 hours post-transfection. The supernatants containing ACE2-GFP-Fc were collected 48 hours post-transfection, filtered through a 0.45-µm membrane (Millex), supplemented with 25 mM HEPES (Gibco), and stored at -80°C. For cell-surface Spike expression, HEK293T cells ($1.125 \times 10^5$) were seeded in poly-L-lysine-coated 48-well plates and transfected the next day with 0.25 µg of Spike plasmid using PEI. The medium was refreshed 8 hours post-transfection. After 24 hours of transfection, cells were incubated with 200 µL of soluble ACE2-GFP-Fc at 4°C in the dark for 2 hours. After incubation, the supernatant was removed, and cells were trypsinized, resuspended in PBS containing 2% paraformaldehyde and 2% FBS, passed through a 70-µm filter, and analyzed using an Attune NxT Flow Cytometer (Invitrogen) to determine the mean fluorescence intensity (MFI) and the percentage of FITC-positive cells. For Spike·F1148 mutants, cells were detached by gentle pipetting instead of trypsinization. To assess antibody-mediated inhibition of Spike-ACE2 binding, Spike-expressing HEK293T cells were incubated with serial dilutions of XG005 nAb [46], anti-HA rabbit mAb, or rabbit IgG at 37°C for 1 hour. After washing, cells were incubated with soluble ACE2-GFP-Fc at 4°C in the dark for 2 hours. Flow cytometry was conducted to calculate the MFI and quantify the percentage of FITC-positive cells. Dose-response curves were generated and analyzed using GraphPad Prism software to calculate the $IC_{50}$ values for each antibody.

## ELISA for quantifying HA epitope accessibility in SC2-VLP^HA

The SARS-CoV-2 (2019-nCoV) Spike RBD ELISA Kit (Sino Biological, KIT40592) was used according to the manufacturer's protocol. The 96-well plate provided in the kit, pre-coated with an anti-Spike RBD mAb, was used to capture SC2-VLPs. Prior to sample loading, the wells were washed twice with 300 µL of 1×Wash Buffer (Sino Biological) for 2 minutes each. A total of 200 µL of SC2-VLPs^HA-containing supernatant was added to each well and incubated at room temperature for 2 hours. After incubation, the wells were washed three times with 300 µL of 1×Wash Buffer (2 minutes per wash). Next, 100 µL of anti-HA mAb (Cell Signaling Technology, #3724), diluted in antibody dilution buffer (Sino Biological), was added and incubated for 1 hour at room temperature. After washing three times with 1×Wash Buffer, 100 µL

of HRP-conjugated goat anti-rabbit secondary antibody (Cell Signaling Technology, #7074; 15 ng/mL) was added and incubated for 1 hour at room temperature. Following three additional washes, 100 µL of Substrate Solution (Sino Biological) was added to each well and incubated in the dark for 20 minutes. The reaction was terminated with 100 µL of Stop Solution (Sino Biological), and the optical density at 450 nm (OD450) was measured using a microplate reader.

## Cell-cell membrane fusion assay

To evaluate Spike-mediated cell-cell membrane fusion and its inhibition, donor cells were generated by transiently transfecting HEK293T cells with plasmids encoding (C-Cfa)-(C-Gluc)-linker-(C-Venus) and Spike at a 1:1 mass ratio. Receptor cells were produced by transiently transfecting HEK293T cells with plasmids encoding (N-Venus)-linker-(N-Gluc)-(N-Cfa), ACE2, and TMPRSS2 at a 1:0.5:0.5 mass ratio. At 24 hours post-transfection, donor and receptor cells were harvested by trypsinization (or pipetting for Spike·F1148 mutants) and co-seeded into 96-well plates at a 1:1 ratio ($3 \times 10^4$ cells per well). After 24 hours of co-culture, fluorescent images were acquired using an inverted fluorescence microscope (Invitrogen, EVOS M5000). Cells were then washed with PBS and lysed with 50 µL of 1 × passive lysis buffer (Promega). Subsequently, 30 µL of the lysate was mixed with 50 µL of *Gaussia* luciferase substrate (Promega), and luciferase activity was measured using a GLOMX luminometer (Promega). For antibody or peptide inhibition assays, donor cells were preincubated at 37°C for 1 hour with serial dilutions of LongHR2_42 peptide [47], anti-HA rabbit mAb, or rabbit IgG, and then mixed 1:1 with receptor cells. Inhibitors were maintained at the same concentration during co-incubation. After 24 hours of co-incubation at 37°C, intracellular *Gaussia* luciferase activity was measured. Dose-response curves were analyzed using GraphPad Prism software to calculate the $IC_{50}$ values for each inhibitor.

## Flow cytometric analysis

HEK293T cells ($1.125 \times 10^5$) were seeded onto poly-L-lysine-coated 48-well plates. After 24 hours, cells were transfected with 0.25 µg of the Spike·NTD[HA] plasmid using PEI transfection reagent (Yeasen, 40816ES01) according to the manufacturer's instructions. The medium was refreshed 8 hours post-transfection. At 24 hours post-transfection, cells were gently resuspended in PBS containing 2% FBS, followed by centrifugation at 1,000 × g for 2 minutes. After two washes with PBS containing 2% FBS, the cells were incubated with anti-HA monoclonal antibody at 4°C for 1 hour. Following two additional washes with PBS containing 2% FBS, the cells were incubated with an FITC-conjugated secondary antibody at 4°C for 1 hour. Subsequently, the cells were washed twice with PBS containing 2% FBS, fixed in PBS containing 2% paraformaldehyde and 2% FBS, and passed through a 70-µm filter. The samples were analyzed on an Attune NxT Flow Cytometer (Invitrogen) to determine the mean fluorescence intensity (MFI) and the percentage of FITC-positive cells.

## Structure analysis

The structure of the SARS-CoV-2 Spike protein was predicted using AlphaFold 3 (https://alphafoldserver.com/), and the structures were visualized with UCSF ChimeraX (https://www.cgl.ucsf.edu/chimerax/).

## Statistical analysis

Statistical analyses were performed using GraphPad Prism software, with specific test details provided in the figure legends.

## Supporting information

**S1 Fig. Optimization of SARS-CoV-2 VLP-based pseudoparticles (SC2-VLPs). (A)** Schematic of experiment design of the packaging and detection of SC2-VLP. SC2-VLP was generated by co-transfecting plasmids expressing Spike (S), membrane (M), envelope (E), nucleocapsid (N) and Fluc-PS9. The SC2-VLP were harvested and used to infect HEK293T

cells co-expressing ACE2 and TMPRSS2. Infectivity was quantified by measuring Firefly luciferase activity 24 hours post-infection. (**B**) Uniform design-based optimization of plasmid transfection ratios. Following the principles of uniform design, different plasmid ratio groups were established for SC2-VLP packaging and HEK293T-ACE2&TMPRSS2 cell infection. Intracellular Firefly luciferase activities were determined at 24 hours post-infection (mean values±SDs, n=3). The initial condition [36] (#1, red) and optimized condition (#7, blue), are indicated. (**C**) Structural protein requirement for SC2-VLP infectivity. The packaged SC2-VLP was used to infect naive HEK293T and HEK293T-ACE2&TMPRSS2 cells, and intracellular Firefly luciferase activities were determined at 24 hours post-infection (mean values±SDs, n=3). (TIF)

**S2 Fig. Characterization of SARS-CoV-2 VLP-based pseudoparticles (SC2-VLPs).** (**A**) Protein composition analysis of SC2-VLPs. Western blotting was performed using methanol-precipitated SC2-VLP supernatants and packaging cell lysates. Blots were probed with indicated antibodies (IB). The values to the left of the blots are molecular sizes in kilodaltons. The representative images shown for each group from multiple independent experiments.(**B**) Visualization of the HA-tag insertion site in the M structure. The HA-tag insertion site is located at the N-terminus of M, as depicted in the column cartoon representation of the unresolved structure (PDB ID: 7VGS). (**C**) Infectivity of SC2-VLP-M$^{HA}$. HEK293T-ACE2/TMPRSS2 cells were infected with SC2-VLP-M$^{HA}$, and intracellular Firefly luciferase activities were determined at 24 hours post-infection (mean values±SDs, n=3). (**D**) Immunoprecipitation of SC2-VLPs-M$^{HA}$. HEK293T cells were transfected with plasmids to package SC2-VLPs-M$^{HA}$, and the supernatant containing SC2-VLPs was collected 3 days post-transfection. Two aliquots were prepared: one was mixed with four volumes of methanol to precipitate the pelleted as input, while the other was incubated with anti-HA beads for protein capture (IP). The captured proteins were subjected to Western blotting assay with the indicated antibodies (IB), with representative images from three biological replicates are shown. The values to the left of the blots are molecular sizes in kilodaltons. (**E**) Structural visualization of the HA-tag insertion sites in the S NTD domain. The image was generated using PDB 6XR8. The enlarged region (boxed) indicated the residues of Q23, I68 and H245 (red) of the NTD domain (light blue). (**F**) The infectivity of HEK293T-ACE2&TMPRSS2 cells by SC2-VLPs-(S-NTD$^{HA}$). The packaged SC2-VLPs-(S-NTD$^{HA}$) was used to infect HEK293T-ACE2&TMPRSS2 cells, and intracellular Firefly luciferase activities were determined at 24 hours post-infection (mean values±SDs, n=3). (**G**) Western blotting of SC2-VLPs-(S-NTD$^{HA}$). The SC2-VLP-containing supernatants precipitated by methanol and the packaging cell lysates were subjected to Western blotting assay with the indicated antibodies (IB). The values to the left of the blots are molecular sizes in kilodaltons. Mock: untransfected HEK293T control. The representative images shown for each group from multiple independent experiments. (TIF)

**S3 Fig. Identification and characterization of key single residue in Mut2 and Mut3 mutants.** (A) The infectivity of SC2-VLP mutants. The packaged SC2-VLPs with single alanine mutations of the Mut2 and Mut3 were used to infect HEK293T-ACE2&TMPRSS2 cells, and intracellular Firefly luciferase activities were determined at 24 hours post-infection (mean values±SDs, n=3, ***$P < 0.001$; two-tailed, unpaired t-test). (B) Structural visualization of the residue C1126 in the prefusion Spike. The enlarged region (boxed) indicated the potential disulfide bond formed between C1126 (red) and C1082 (blue). (C) The infectivity of SC2-VLPs with the C1126A and C1082A mutants. The packaged SC2-VLPs-(Spike·C1126A) and SC2-VLPs-(Spike·C1082A) were used to infect HEK293T-ACE2&TMPRSS2 cells, and intracellular Firefly luciferase , i.e.,s were determined at 24 hours post-infection (mean values±SDs, n=3). (D) Western blotting of SC2-VLPs with the C1126A and C1082A mutants. The SC2-VLP-containing supernatants precipitated by methanol and the packaging cell lysates were subjected to Western blotting assay with the indicated antibodies (IB). The values to the left of the blots are molecular sizes in kilodaltons. The representative images shown for each group from multiple independent experiments. (TIF)

**S4 Fig. The alignment of PFPH-1 and 3H/CH across the betacoronaviruses.** (A) The structural alignments of PFPH-1, 3H and CH across the betacoronaviruses. The image was generated by aligning PFPH-1, 3H, and CH domains of the SARS-CoV-2 (red, PDB 8FDW) with SARS-CoV (blue, PDB 6M3W) and MHV (green, PDB 6B3O), which yielded a root mean square deviations (RMSDs) of 0.804 Å (SARS-CoV-2 with SARS-CoV), 0.589 Å (SARS-CoV-2 with MHV) and 0.649 Å (SARS-CoV with MHV), respectively. The enlarged box indicated the key hydrophobic residues. (B-C) The amino acid sequence alignments of 3H (panel B) and CH (panel C) were performed using Clustal Omega and ESPript 3.0. Red highlighting represents 100% identity, whereas blue boxed red fonts shows a global score of 70% identity based on ESPript 3.0 parameters. Brown boxes indicate the key hydrophobic residues.
(TIF)

**S5 Fig. Cell surface staining of Spike-NTD^HA in F1148 Mutants.** (**A**) Schematic representation experimental design of the cell surface staining of Spike-NTD^HA. Spike-NTD^HA was expressed on the surface of HEK293T cells, followed by incubation with primary antibody (Anti-HA mAb) and secondary antibody at 4°C for 1 hours. The percentage of FITC-positive cells and the mean fluorescence intensity (MFI) was determined by flow cytometry. (**B**) The population of NTD^HA-staining-positive cells was determined by normalizing the percentage of FITC-positive cells and the mean fluorescence intensity (MFI) of the Spike.F1148 mutants to that of WT (mean values ± SDs, n = 3).
(TIF)

**S6 Fig. Establishment and characterization of cell-cell membrane fusion.** (**A**) Quantitative analysis of Spike-mediated membrane fusion efficiency. The donor cell population expressing Spike was incubated with different types of recipient cell populations at 37°C, and the intracellular Gluc luciferase activities were determined at 24 hours post-incubation. (mean values ± SDs, n = 3). (**B**) The fluorescence images of panel A. Incubate the donor cell population expressing Spike with the recipient cell population expressing ACE2 at 37°C, fluorescence images were acquired at 24 hours post-incubation.
(TIF)

**S7 Fig. Characterization of the 3H/CH hydrophobic lock (hydrolock) interaction.** (**A-C**) Schematic of the "hydrolock interaction" model. The images were generated using PDB 6XR8, PDB 8Z7P and PDB 8FDW, illustrating the overview (i), L977 (ii) and F782 (iii) at different conformational stages (A, prefusion; B, E-FIC, early fusion intermediate conformation. C, postfusion). The 3H (blue), CH (magenta), HR1 (yellow), FP (red), CD (orange), helix976–984 (khaki) and loop776–785 (green) are shown. The khaki and green triangular arrows indicate the residues L977 and F782, respectively.
(TIF)

**S8 Fig. Cell surface staining of Spike-NTD^HA in L977G and F782G Mutants.** The population of NTD^HA-staining-positive cells was determined by normalizing the percentage of FITC-positive cells and the mean fluorescence intensity (MFI) of the Spike mutants to that of WT (mean values ± SDs, n = 3).
(TIF)

**S9 Fig. Characterization of exogenous HA linear epitope replacement in SC2-VLPs.** (**A**) The infectivity of SC2-VLP with Spike^HA linear epitope replacement mutants. The packaged SC2-VLPs-(S-SH^HA) were used to infected HEK293T-ACE2&TMPRSS2 cells, and intracellular Firefly luciferase activities were determined at 24 hours post-infection (mean values ± SDs, n = 3). The black dashed line represents the baseline for detectable infection signals, which is applicable for subsequent neutralization assays. (**B**) Western blotting of SC2-VLPs-(S-SH^HA). The SC2-VLP-containing supernatants precipitated by methanol and the packaging cell lysates were subjected to Western blotting assay with the indicated antibodies (IB). The values to the left of the blots are molecular sizes in kilodaltons. Mock, untransfected HEK293T control.
(TIF)

**S10 Fig. Negative control for IgG in neutralization assay.** The infectivity of SC2-VLP-(S-SH<sup>HA</sup>) after incubation with a 5-fold concentration gradient of rabbit IgG at 37°C for 1 hour. The intracellular Firefly luciferase activities were determined at 24 hours post-infection. The relative infectivity was determined by normalizing the luciferase activity of SC2-VLPs incubated with IgG to that of SC2-VLPs incubated with medium as a control. The curve fitting was performed using GraphPad Prism software. (mean values ± SDs, n = 3).
(TIF)

**S11 Fig. Antibody accessibility of regions proximal to PFPH-1 and PFPH-3.** (**A**) Experimental schematic for assessing SH region antibody accessibility using ELISA. Anti-Spike RBD mAb-coated 96-well plates were used to capture SC2-VLPs-(S-SH<sup>HA</sup>). Sequential incubations were performed by adding the primary antibody (anti-HA mAb), followed by an HRP-conjugated secondary antibody. A chemiluminescent substrate was added to detect the signal. (**B**) Quantitative analysis of SH accessibility using OD450 measurements. A 5-fold concentration gradient of anti-HA mAb was applied, and the OD450 values were measured to reflect the accessibility of different regions of SH. (mean values ± SDs, n = 3).
(TIF)

**S12 Fig. Western blotting of Spike· F1148 mutants.** The plasmids Spike·WT and Spike·F1148 mutants were transfected into HEK293T cells. After 24 hours, cell lysates were harvested and subjected to Western blotting assay with the indicated antibodies (IB). The values to the left of the blots are molecular sizes in kilodaltons. Mock: naive HEK293T cells without plasmid transfection.
(TIF)

**S1 Table. Summary of Mutations and Abbreviations Used in Plasmids Described in the Manuscript.**
(XLSX)

**S2 Table. Plasmid Transfection Ratios Used for Uniform Experimental Design.**
(XLSX)

**S3 Table. Raw data required to replicate the results of this manuscript.**
(XLSX)

**S1 Data. AlphaFold 3-based structural prediction of the Spike protein trimer encompassing residues 1135–1273.**
(CIF)

**S1 Text. Construct sequences used in the Spike–ACE2 binding and membrane fusion assays.**
(DOC)

## Acknowledgments

We would like to thank Yungang He for his help in setting parameters for uniform design.

## Author contributions

**Conceptualization:** Zhigang Yi.

**Investigation:** Fuzhi Lei, Yahan Lei.

**Methodology:** Fuzhi Lei, Zhigang Yi.

**Project administration:** Zhenghong Yuan.

**Resources:** Zhenghong Yuan, Zhigang Yi.

**Supervision:** Zhenghong Yuan, Zhigang Yi.

**Validation:** Fuzhi Lei, Yahan Lei.

**Visualization:** Fuzhi Lei.

**Writing – original draft:** Fuzhi Lei.

**Writing – review & editing:** Zhigang Yi.

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
