## [Decision Letter · Decision Letter 0]

17 Jun 2025

PPATHOGENS-D-25-01115

The F1148 Hydrophobic Lock: A Critical Determinant of SARS-CoV-2 Spike Protein-Mediated Membrane Fusion via the 3H/CH Cavity

PLOS Pathogens

Dear Dr. Yi,

Thank you for submitting your manuscript to PLOS Pathogens. After careful consideration, we feel that it has merit but does not fully meet PLOS Pathogens's publication criteria as it currently stands. Therefore, we invite you to submit a revised version of the manuscript that addresses the points raised during the review process.

Please submit your revised manuscript within 60 days Aug 16 2025 11:59PM. If you will need more time than this to complete your revisions, please reply to this message or contact the journal office at plospathogens@plos.org. Please include the following items when submitting your revised manuscript:

We look forward to receiving your revised manuscript.

Kind regards,

Tongqing Zhou, Ph.D.

Guest Editor

PLOS Pathogens

Sonja Best

Section Editor

PLOS Pathogens

 Sumita Bhaduri-McIntosh

Editor-in-Chief

PLOS Pathogens

orcid.org/0000-0003-2946-9497

 Michael Malim

Editor-in-Chief

PLOS Pathogens

orcid.org/0000-0002-7699-2064

**Journal Requirements:**

At this stage, the following Authors/Authors require contributions: Fuzhi Lei, Yahan Lei, Zhenghong Yuan, and Zhigang Yi. Please ensure that the full contributions of each author are acknowledged in the "Add/Edit/Remove Authors" section of our submission form.

2) We note that your Supplementary Figures are uploaded twice in the online submission form. Please remove any unnecessary files from your revision, and make sure that only those relevant to the current version of the manuscript are included.

3) We noticed that you used the phrase 'data not shown' in the manuscript. We do not allow these references, as the PLOS data access policy requires that all data be either published with the manuscript or made available in a publicly accessible database. Please amend the supplementary material to include the referenced data or remove the references.

4) We do not publish any copyright or trademark symbols that usually accompany proprietary names, eg ©,  ®, or TM  (e.g. next to drug or reagent names). Therefore please remove all instances of trademark/copyright symbols throughout the text, including:

- TM on page: 32.

5) Please upload all main figures as separate Figure files in .tif or .eps format. For more information about how to convert and format your figure files please see our guidelines: 

6) We have noticed that you have uploaded Supporting Information files, but you have not included a list of legends. Please add a full list of legends for your Supporting Information files after the references list.

7) We notice that your supplementary figures are uploaded with the file type 'Figure'. Please amend the file type to 'Supporting Information'. Please ensure that each Supporting Information file has a legend listed in the manuscript after the references list.

8) Some material included in your submission may be copyrighted. According to PLOSu2019s copyright policy, authors who use figures or other material (e.g., graphics, clipart, maps) from another author or copyright holder must demonstrate or obtain permission to publish this material under the Creative Commons Attribution 4.0 International (CC BY 4.0) License used by PLOS journals. Please closely review the details of PLOSu2019s copyright requirements here: PLOS Licenses and Copyright. If you need to request permissions from a copyright holder, you may use PLOS's Copyright Content Permission form.

Potential Copyright Issues:

i) Figures 2A, 3I, 3G, 6A, and S1A. Please confirm whether you drew the images / clip-art within the figure panels by hand. If you did not draw the images, please provide (a) a link to the source of the images or icons and their license / terms of use; or (b) written permission from the copyright holder to publish the images or icons under our CC BY 4.0 license. Alternatively, you may replace the images with open source alternatives. See these open source resources you may use to replace images / clip-art:

9) In the online submission form, you indicated that "The AlphaFold 3-predicted Spike SH structures used in this study are available upon request." All PLOS journals now require all data underlying the findings described in their manuscript to be freely available to other researchers, either

1. In a public repository

2. Within the manuscript itself

3. Uploaded as supplementary information.

10) Please amend your detailed Financial Disclosure statement. This is published with the article. It must therefore be completed in full sentences and contain the exact wording you wish to be published.

3) If any authors received a salary from any of your funders, please state which authors and which funders.

11) Please ensure that the funders and grant numbers match between the Financial Disclosure field and the Funding Information tab in your submission form. Note that the funders must be provided in the same order in both places as well.

12) Please provide a completed 'Competing Interests' statement, including any COIs declared by your co-authors. If you have no competing interests to declare, please state "The authors have declared that no competing interests exist". 

**Reviewers' Comments:**

Reviewer's Responses to Questions

**Part I - Summary**

Reviewer #1: This manuscript titled “The F1148 Hydrophobic Lock: A Critical Determinant of SARS-CoV-2 Spike Protein-Mediated Membrane Fusion via the 3H/CH Cavity” presents a systematic investigation of the hydrophobic interaction centered on residue F1148 in the spike stem helix (SH) region. For clss-I viral fusion proteins, the HR1/HR2 6-helix bundle is the core structure of membrane fusion, while the critical domains or residues that drive this process are still unclear. In this study, the authors characterize a “hydrolock” interaction formed by F1148 and the 3H/CH cavity as an essential structural and functional element for spike-mediated membrane fusion. Overall, this study is novel and interesting. However, there are some concerns in this study.

Reviewer #2: In this work, Lei and colleagues performed systematic mutagenesis within a conserved region of SARS-CoV-2 spike protein, the stem-helix (SH) on S2 subunit, to identify key residues involved in viral infection and membrane fusion. The results highlighted helices preserved in the postfusion spike, which the authors termed as postfusion-preserved helices (PFPH1, PFPH2 and PFPH3). Through structural analyses of the spike protein in prefusion, early-fusion intermediate and postfusion states, the authors identified a hydrophobic pocket – referred to as the “hydrolock” -- which accommodates hydrophobic residues L977, F782 and F1148 in a state-dependent manner. This pocket interacts with these residues alternately in prefusion, early-fusion intermediate and postfusion states. Mutating these residues to hydrophilic amino acids or disrupting the hydrophobic pocket significantly impaired spike-mediated viral infectivity. Moreover, neutralizing antibodies targeting the stem-helix region were found to bind the spike and clash with its postfusion state. To further dissect the neutralizing mechanism of these antibodies, the authors employed HA replacement mutagenesis to map critical regions within the stem-helix domain. They found that antibody binding to regions adjacent to PFPHs is essential for blocking membrane fusion. In summary, the study provides a detailed structural comparison of available spike models at different stages during fusion and identifies a conserved hydrophobic cavity that is critical for infectivity, likely through facilitating S2 refolding. Given the conservation of this region across coronaviruses, it represents a promising target for the development of broad-spectrum antiviral therapies. This work is well-performed and incorporates multiple assays – including infectivity, ACE2 binding, membrane-membrane fusion and epitope exposure for neutralization -- to functionally characterize spike mutants. The systematic mutagenesis of the stem-helix region provides valuable insights for pan-coronavirus vaccine and therapeutic development.

Reviewer #3: In this paper, the authors find that the hydrophobic interaction between F782/FF148 and 3H/CH cavity and also the postfusion-preserved helice-1 (PFPH-1)/PFPH-2 are important for conformational change of coronavirus spike protein and virus entry. Based on this research, they hypothesize a novel model of coronavirus entry. Overall, this is a good paper which can provide us with some new targets for antiviral drug and coronavirus vaccine design. However, there are some questions remaining to be addressed, please see the major and minor issues.

**Part II – Major Issues: Key Experiments Required for Acceptance**

Reviewer #1: 1. The F1148 hydrophilic mutants nearly abolished the virus entry (Fig. 3D) and severely impaired the membrane fusion efficiency (Fig. 3J) but still maintained more than 50% of ACE2-binding efficiency (Fig. 3H). The ACE2-binding ability is retained, which only means the RBD conformation can be maintained to some extent, but the overall prefusion conformation may still be changed, which could be the reason for the ability abolishment of virus entry and membrane fusion. A panel of antibodies targeting different regions of spike protein are needed to confirm the prefusion conformation like using flow cytometry.

2. The fusion signals of spike/ACE2 groups seem to be significantly different in Fig. S5A and S5C. In Fig. S5C, the TMPRSS2 group should be assembled into Fig. S5A and also S5B as one data set and compare them in parallel.

3. In line 304, p16 and Fig. 4B-C, the authors generated hydrophilic substitutions (L977G and F782G) and assessed the impact on virus entry. It is risky since the smallest glycine mutation is commonly used to increase flexibility or disrupt the secondary structure. The authors should assess the overall prefusion conformation using different antibodies in flow cytometry, not only ACE2 binding.

Reviewer #2: The authors detected secreted proteins in the supernatant using methanol precipitation, intending to represent viral proteins incorporated into VLPs. However, this method may non-selectively precipitate all secreted proteins. Since the spike sequence used in this work is from the original strain which encodes one of the least stable spike proteins, the observed S1 signal in the supernatant could result from S1 shedding from the cellular plasma membrane rather than VLP-incorporated spike. Thus, conclusions about spike assembly defects based solely on WB results of HA-tagged S1 are not sufficiently supported. The authors should also present WB data for S2 in parallel, as the levels of S1 and S2 do not always correlate, e.g., see Figure S1G and S7B. Notably, in Figure S7B, the signal of S1 was detected in the supernatant for 1205-S-SHHA and 1203-S-SHHA, while S2 was absent, strongly suggesting S1 shedding.

Similarly, the hACE2-GFP-Fc binding assay, performed on cell surface, should be normalized to the amount of S2 on the surface to exclude the effect of S1 shedding. Without addressing this issue, the phenotypic interpretations of spike mutants across different assays may need to be reconsidered. For example, although F782 is proposed to engage the hydrophobic cavity in the early-fusion intermediate, the F782G mutation severely disrupted spike assembly on VLP, prior to formation of the intermediate. Likewise, the L977G mutation nearly abolished VLP infectivity while remaining functional in a cell-cell fusion assay. These discrepancies require further clarification.

Reviewer #3: 1. Line 244, have you verified in the soluble SARS-CoV-2 S2P spike to see if the molecular weight is increased by F1148I ?

2. For the Spike-ACE2 binding assay (lines 261-263, lines 668-669), trypsin may decrease the amount of protein or destroy the structure the protein on cell surface which may affect the result of the assay. For related result, I think it’s better to use EDTA to digest the cells. And for the binding ability, it’s better to calculate MFI with different antibody concentrations, not percentage of positive cells with a single antibody dose.

3. Line 498, in figure 3H, compared with WT, the hydrophilic mutations can lead to ~50% decrease, I think the change is large, maybe for the mutations, the binding affinity is not enough to trigger the following conformational change, so the fusion is decreased in figure 3J, maybe not due to the deficiency of interaction between F1148 and 3H/CH cavity, do you think it is possible?

**Part III – Minor Issues: Editorial and Data Presentation Modifications**

Reviewer #1: 1. In line 144, p8, “residues 1135-1273” were used for Alphafold prediction, which is not consistent with “AF3 prediction” in Figure 1C. The prediction region and real model 6XR8 should have some overlap.

2. In line 147, p9, the alignment yielded a RMSD value, please specify the residue range used for the alignment.

3. In line 179-180, p10, please carefully check the description “as assessed by IB:S2”, which appears two times here. Please correct it accordingly.

4. In Figure 3D, please indicate whether the real density or surface display mode is shown for the model.

5. In line 229, the residue F1148 is not strictly conserved across Coronaviridae.

6. Since F1148I mutant exhibited a higher molecular weight than that of WT (Fig. 3E), did the authors check the whole coding sequence for this construct, even the complete plasmid sequence?

7. The properties of amino acids are well known in Fig. 3F, which should be removed or put into supplementary.

8. Since Mut 22 also reduced the VLP infectivity by over 10-fold, the data for Mut 22 should also be included in Figure 2D.

9. A careful language edit is recommended in this manuscript.

Reviewer #2: 1. The address of the author Yahan Lei is missing.

2. Lines 139-141: “The resolved structure of prefusion and … CT domains” is an incomplete sentence and should be revised.

3. Line 152: the correct notation should be “PFPH3 (1179-1193aa)”.

4. In Figure 1A, please indicate the residue numbers for each domain.

5. In Figure 1B, the HR2 helix is predicted by AlphaFold rather than derived from experimental data. Please note this in the figure. Also, consider placing Figure 1B after Figure 1C to improve logic flow.

6. In Figure 1C, have you attempted to predict the S2 structure using AlphaFold3? Given the stability of the postfusion state, this may directly yield a postfusion spike.

7. In figure 2C, please apply a consistent color scheme as used in other figures.

8. Why is mutant “mut22” missing in Figure 2D?

9. In Figure 4K, the results are incomplete. Please at least show signals of S1, S2, N, Actin.

10. In Figure 5B, it should be labeled “PFPH-1”.

11. In Figure 7, it should be “Late stage” rather than “Later stage”.

12. In Supplementary Fig.1, please specify the conditions in #1 - #22, such as the ratio of the transfected components. The phrase “a uniform design strategy” is too vague. This information would benefit other researchers working on VLP optimization.

13. Supplementary Fig. 2B-D appear unrelated to the core findings of the paper, though their inclusion is not problematic.

14. In Supplementary. Fig. 5, Gluc signal intensity varies considerably between panels A and C. In panel A, the fold change between positive and negative controls exceeds 100-fold, while in C panel it is less than 10-fold. This inconsistency should be addressed or explained.

Reviewer #3: 1. The stem helix may be only conserved among the beta-coronaviruses (line 42), not all the coronaviruses. And I have checked some spike sequences of different coronaviruses, the major hydrophobic residues F1148 you found in this paper also seems only conserved in beta-coronaviruses. So, for the sake of accuracy, I think you need to check the whole manuscript and change the description, your findings in this job may be only suited to beta-coronaviruses, not all the coronaviruses.

2. Although the writing is generally clear, there are some places which need to be corrected by a native English speaker, for example lines 47-49, 105-109 and so on. Some sentences are difficult to understand.

3. Why do you define the residues 1141-1211 as stem helix? As reported in most of the published papers, the stem helix only contains residues 1139-1163, and residues 1139-1162 is available in prefusion conformation of SARS-CoV-2 spike (PDB ID: 6XR8).

4. Line 151-152, typo, should be PFPH-1, PFPH1-2 and PFPH-3, “-3” is lost.

5. For sequence alignment (fig 1B), “.” is always used to indicate the similar amino acids, “-” for the deficiency. And for all the panels of sequence alignment in the paper, through the numbers above the sequences, I cannot find the residues as you described in the main text. For example, F1148 cannot be found easily in figure 1B.

6. Line 571-572, 0.096 μg + 0.1952 μg + 0.926 μg + 1.12 μg do not equal to 4 μg, please check. And some of the methods are described too simply. For example, for the method of western blotting (line 599) and description in lines 178-181, through these sentences, I know you can use the anti-HA antibody to detect S1 with HA insertion, but I felt a little confused that how you could detect the S2. After I checked the “Antibodies and peptide” (line 581), I think you may use this anti-S2 antibody (GeneTex, GTX632604) to conduct the WB. The similar question for figure S7B, before I found the RBD antibody (GeneTex, GTX635692) in the “Antibodies and peptide” section, I also felt a little confused on how to detect RBD without HA insertion. As I remember, most RBD antibodies recognize spatial conformation, not linear epitopes, may be not used to do WB. I think you may need to describe directly in the method, use which antibody to detect which fragment, it may be easier to be understood for the readers. Not only the WB methods, please check all the other methods to make sure that it’s more detailed.

7. Line 217, “S1147, F1148, and K1149” may be better.

8. Figure 3J & H, dose F1148W and F1148M affect the binding and fusion?

9. Line 315, it’s better to put the data in the supplemental figures.

10. The names of Y-axis in fig 3J and fig 4B are different, I think should be the same name.

11. In figure S7B, for some channels, why RBD can be detected, S2 with HA insertion can not?

PLOS authors have the option to publish the peer review history of their article (what does this mean? ). If published, this will include your full peer review and any attached files.

**Do you want your identity to be public for this peer review?** For information about this choice, including consent withdrawal, please see our Privacy Policy .

Reviewer #1: No

Reviewer #2: No

Reviewer #3: No

**Figure resubmission:**
---

## [Decision Letter · Decision Letter 1]

9 Sep 2025

Dear Dr Yi,

We are pleased to inform you that your manuscript 'The F1148 Hydrophobic Lock: A Critical Determinant of SARS-CoV-2 Spike Protein-Mediated Membrane Fusion via the 3H/CH Cavity' has been provisionally accepted for publication in PLOS Pathogens.

Before your manuscript can be formally accepted you will need to (1) make  minor revisions of the manuscript based on Reviewer 2's comments,(2) complete some formatting changes, which you will receive in a follow up email. A member of our team will be in touch with a set of requests.

Best regards,

Tongqing Zhou, Ph.D.

Guest Editor

PLOS Pathogens

Sonja Best

Section Editor

PLOS Pathogens

Sumita Bhaduri-McIntosh

Editor-in-Chief

PLOS Pathogens

orcid.org/0000-0003-2946-9497

Michael Malim

Editor-in-Chief

PLOS Pathogens

orcid.org/0000-0002-7699-2064

Reviewer #1:

Reviewer #2:

Reviewer #3:

Reviewer Comments (if any, and for reference):

Reviewer's Responses to Questions

**Part I - Summary**

Reviewer #1: Overall, the authors have addressed my major concerns.

Reviewer #2: The authors have addressed most of my concerns. They demonstrated the unsuccessful detection of S2 mutations with the currently available anti-S2 antibodies, which unfortunately limits the conclusions regarding spike protein packaging, as undetected or reduced S1 signals in WB alone cannot distinguish between a defect in spike packaging and instability of spike heterodimerization. Including this limitation in the discussion would be sufficient, especially since this issue is not the primary focus of the manuscript.

Reviewer #3: In this paper, the authors find that the hydrophobic interaction between F782/FF148 and 3H/CH cavity and also the postfusion-preserved helice-1 (PFPH-1)/PFPH-2 are important for conformational change of betacoronavirus spike protein and virus entry. Based on this research, they hypothesize a novel model of betacoronavirus entry. After revision, I think this is a good paper which can provide us with some new targets for antiviral drug and betacoronavirus vaccine design.

**Part II – Major Issues: Key Experiments Required for Acceptance**

Reviewer #1: (No Response)

Reviewer #2: None

Reviewer #3: I do not have any major issues now. The authors have addressed my concerns.

**Part III – Minor Issues: Editorial and Data Presentation Modifications**

Reviewer #1: (No Response)

Reviewer #2: There are several typos when citing supplementary figures, such as “S1A Fig” in Line 161, etc.

Reviewer #3: I think it is better to include the figures in the rebuttal letter into the supplementary materials. Some readers may have the same concerns as the reviewers without these figures.

PLOS authors have the option to publish the peer review history of their article (what does this mean? ). If published, this will include your full peer review and any attached files.

**Do you want your identity to be public for this peer review?** For information about this choice, including consent withdrawal, please see our Privacy Policy .

Reviewer #1: No

Reviewer #2: No

Reviewer #3: No

---

## [Editor Report · Acceptance letter]

Dear Dr Yi,

We are delighted to inform you that your manuscript, " 

The F1148 Hydrophobic Lock: A Critical Determinant of SARS-CoV-2 Spike Protein-Mediated Membrane Fusion via the 3H/CH Cavity," has been formally accepted for publication in PLOS Pathogens.

Best regards,

Sumita Bhaduri-McIntosh

Editor-in-Chief

PLOS Pathogens

orcid.org/0000-0003-2946-9497

Michael Malim

Editor-in-Chief

PLOS Pathogens

orcid.org/0000-0002-7699-2064